# Germline/soma distinction in *Drosophila* embryos requires regulators of zygotic genome activation

Megan M Colonnetta, Paul Schedl*, Girish Deshpande*

Department of Molecular Biology, Princeton University, Princeton, United States

**Abstract** In *Drosophila melanogaster* embryos, somatic versus germline identity is the first cell fate decision. Zygotic genome activation (ZGA) orchestrates regionalized gene expression, imparting specific identity on somatic cells. ZGA begins with a minor wave that commences at nuclear cycle (NC)8 under the guidance of chromatin accessibility factors (Zelda, CLAMP, GAF), followed by the major wave during NC14. By contrast, primordial germ cell (PGC) specification requires maternally deposited and posteriorly anchored germline determinants. This is accomplished by a centrosome coordinated release and sequestration of germ plasm during the precocious cellularization of PGCs in NC10. Here, we report a novel requirement for Zelda and CLAMP during the establishment of the germline/soma distinction. When their activity is compromised, PGC determinants are not properly sequestered, and specification is disrupted. Conversely, the spreading of PGC determinants from the posterior pole adversely influences transcription in the neighboring somatic nuclei. These reciprocal aberrations can be correlated with defects in centrosome duplication/separation that are known to induce inappropriate transmission of the germ plasm. Interestingly, consistent with the ability of bone morphogenetic protein (BMP) signaling to influence specification of embryonic PGCs, reduction in the transcript levels of a BMP family ligand, *decapentaplegic* (*dpp*), is exacerbated at the posterior pole.

*For correspondence:
pschedl@princeton.edu (PS);
gdeshpan@princeton.edu (GD)

**Competing interest:** The authors declare that no competing interests exist.

## Editor's evaluation

The early differentiation of germ cells, those that will form egg and sperm, is a critical and nearly universal step in animal development. This paper reveals new layers of molecular and cellular regulation that control this process in the fly, and as such be of broad interest to cell and developmental biologists, especially those interested in critical cell fate decisions. The paper contains a wealth of experimental data demonstrating that processes generally thought to be restricted to somatic cells alter the differentiation of germ cells, but provides only limited functional interpretation of the observed phenotypes.

## Introduction

Embryogenesis in *Drosophila melanogaster* begins with a series of synchronous nuclear divisions that proceed in a shared cytoplasm or syncytium. While the first eight nuclear cycles (NCs) take place in the central region of the embryo (pre-blastoderm), nuclei begin to migrate toward the periphery during NC9 to initiate formation of the syncytial blastoderm (*Farrell and O'Farrell, 2014*). It is at this juncture in development that the first cell fate decision is made, namely germline versus soma. As the nuclei begin migrating outward from the center of the embryo, a select few move ahead of the rest into the posterior pole where they induce the formation of pole buds. These nuclei are destined to form the germline of the embryo, while the remaining nuclei assume somatic fate. The posterior pole is special

due to the presence of germ plasm enriched in maternally deposited germline determinants. The germline determinants are assembled at the posterior of the oocyte during the last stages of oogenesis. This process is orchestrated by *oskar* (*osk*) which recruits all the other germ plasm constituents, including Vasa protein and *nanos (nos), germ cell-less (gcl),* and *polar granule component (pgc)* mRNAs (*Ephrussi and Lehmann, 1992*). The germ plasm is released from the posterior cortex upon the entry of centrosomes associated with the incoming nuclei. The germ plasm is sequestered in the pole buds which subsequently undergo cellularization to form primordial germ cells (PGCs). This segregation of germline determinants in newly formed cells occurs via trafficking on centrosome nucleated microtubules (*Lerit and Gavis, 2011*; *Raff and Glover, 1989*). As a result, PGCs receive the full complement of the germ plasm, and the surrounding somatic nuclei are protected from being exposed to it. Once cellularization is complete, the newly formed PGCs have special properties that distinguish them from the surrounding soma including transcriptional quiescence, specialized global chromatin architecture, and limited mitotic self-renewal (*Deshpande et al., 2004*; *Ephrussi and Lehmann, 1992*; *Lebedeva et al., 2018*; *Marlow, 2015*; *Santos and Lehmann, 2004*; *Su et al., 1998*).

While the newly formed PGCs are exiting the cell cycle and shutting down transcription, the surrounding somatic nuclei continue rapid, synchronous nuclear division cycles and activate zygotic transcription. Zygotic genome activation (ZGA) commences even prior to the migration of nuclei to the periphery of the embryo when transcription of small subset of genes can be detected (*Ali-Murthy et al., 2013*; *Hamm and Harrison, 2018*; *Schulz and Harrison, 2019*). There is a modest yet discernible increase in the overall level of zygotic transcription after the nuclei reach the surface of the embryo, and this minor wave of ZGA continues through NC13 to the beginning of NC14, when transcription is substantially upregulated. During the minor wave, several critical developmental events occur in the soma in a temporally coordinated manner, including not only PGC formation but also somatic sex determination and the establishment of the initial body plan of the embryo (*Hamm and Harrison, 2018*; *Marlow, 2015*; *Salz and Erickson, 2010*; *Santos and Lehmann, 2004*; *Tadros and Lipshitz, 2009*). At approximately NC11, the master determinant of female fate, *Sex lethal* (*Sxl*) commences transcription from its establishment promoter, *Sxl-Pe*, prompted by X chromosome counting elements transcribed starting at NC6 (*Estes et al., 1995*; *Keyes et al., 1992*; *Salz and Erickson, 2010*). Subsequently, the transcription of patterning genes is gradually activated to establish the anterior-posterior and dorsal ventral body axis (*St Johnston and Nüsslein-Volhard, 1992*; *Schroeder et al., 2004*).

The global activation of transcription in *Drosophila* depends on several factors that act genome-wide to coordinate ZGA. The known factors include Zelda (Zld), CLAMP, and GAGA factor (GAF) (*Bhat et al., 1996*; *Colonnetta et al., 2021a*; *Duan et al., 2021*; *Gaskill et al., 2021*; *Harrison et al., 2011*; *Liang et al., 2008*; *McDaniel et al., 2019*; *Moshe and Kaplan, 2017*; *Nien et al., 2011*). Among these, by far the best studied is Zld, a $C_2H_2$ zinc-finger transcription factor that binds to target sequences throughout the genome. *zld* RNAs are maternally deposited and are rapidly translated following fertilization. Zld is thought to be a 'pioneer' factor capable of establishing regions of open chromatin that permit the recruitment of transcription factors (*Blythe and Wieschaus, 2016*; *Fernandez Garcia et al., 2019*; *Schulz et al., 2015*; *Sun et al., 2015*). Unlike canonical transcriptional regulators that control the expression of only a few genes, Zld impacts expression genome-wide by shaping chromatin architecture (*Blythe and Wieschaus, 2016*; *Hug et al., 2017*; *Schulz et al., 2015*). While the function of Zld during the major wave of ZGA at NC14 has been the focus of much study, it is required continuously throughout both the minor and major waves of ZGA (*McDaniel et al., 2019*) and binds to a subset of its target genes at the onset of the minor wave (NC8) itself (*Harrison et al., 2011*; *Nien et al., 2011*). Moreover, it has been shown that maternal deposition and early functions of Zld are vital for successful embryogenesis (*Hamm et al., 2017*). For instance, during the minor wave, Zld activates a small number of genes whose products are involved in determining somatic sex (based on counting of X chromosomes) and initial body plan patterning (e.g. early gap genes) (*Liang et al., 2008*; *Nien et al., 2011*; *ten Bosch et al., 2006*).

Though the level of transcription during the minor wave of ZGA is relatively modest, curiously, among early Zld targets are patterning genes such as *engrailed* and *even-skipped* (*eve*) (*Ali-Murthy et al., 2013*). Furthermore, it has been suggested that zygotic transcription in pre-syncytial blastoderm could be crucial for the establishment of early topologically associated domains that determine the overall transcriptional landscape of a developing embryo (*Hug et al., 2017*). Taken together, these observations prompted us to examine possible unanticipated function(s) of the minor wave of ZGA.

**Table 1.** Primordial germ cell (PGC) counts decrease with *zld* or *clamp* knockdown.

| Genotype | Mean PGCs | SD | N |
|---|---|---|---|
| *egfpi*[m] | 19.2 | 3.67 | 12 |
| *zldi*[1m] | 13.7 | 6.13 | 15 |
| *egfpi*[z] | 19.7 | 1.95 | 10 |
| *zldi*[2z] | 12.9 | 3.94 | 8 |
| *clampi*[z] | 12.6 | 5.48 | 8 |

The online version of this article includes the following source data for table 1:

**Source data 1.** Raw data summarized in *Table 1*: Primordial germ cell (PGC) counts for individual embryos.

Traditional models posit that *Drosophila* germline fate is determined in an entirely cell autonomous manner ('preformation') by maternally supplied germline determinants that are localized at the posterior pole of the embryo (*Extavour and Akam, 2003*; *Strome and Updike, 2015*). However, recent studies have shown that the proper specification of PGC identity in flies is not exclusively determined by preformation-based mechanism. Instead, it also depends upon extrinsic somatic signals from the BMP pathway (bone morphogenetic protein pathway) ligand Dpp (Decapentaplegic) which is expressed in the early embryo (*Colonnetta et al., 2022*). For these reasons, we wondered whether the chromatin factors that set the stage for the major wave of ZGA in NC14 also impact the process of PGC specification in early blastoderm embryos. Making this idea even more plausible is the fact that compromising ZGA regulators adversely influences nuclear division, centrosomes, and cytoskeletal elements (*Bhat et al., 1996*; *Colonnetta et al., 2021a*; *Liang et al., 2008*; *Staudt et al., 2006*), all of which play a critical role in PGC formation and specification. Here, we have examined the role of two different ZGA regulators, Zld and CLAMP, during the establishment of germline/soma distinction in early embryos, and we demonstrate their requirement for proper specification of both soma and PGCs.

## Results

### Embryos compromised for ZGA regulators have reduced PGC counts

Previous studies on the ZGA regulator Zld suggested that PGC formation is relatively normal in embryos derived from females carrying *zld*[-] germline clones; however, these studies only showed that PGCs are present (*Liang et al., 2008*). To determine if compromising *zld* has any undetected effect on nascent germline formation and/or specification, we counted the number of PGCs in embryos maternally compromised for *zld*. For this purpose, we used a germline driver, *maternal-tubulin-GAL4* (or *mat-Gal4*, i.e. *67.15*), to drive expression of a *zld* shRNA transgene, *pUAS-zld* (*zldi*[1m]) in the mother. This RNAi knockdown strategy has been shown to recapitulate the phenotypic consequences observed in *zld*[-] germline clone embryos (*Sun et al., 2015*; *Yamada et al., 2019*). As indicated in *Table 1*, we observe a small but significant decrease in the number of PGCs in syncytial and cellular blastoderm stage *zldi*[1m] embryos. While control embryos (*egfpi*[m]) derived from mothers expressing an shRNA directed against *egfp* had 19.2 PGCs on average (n=12), *zldi*[1m] embryos had an average of 13.7 PGCs (n=15; p=0.021 by t-test) (*Table 1*).

It remains possible that the maternal RNAi-based knockdown could also impact zygotic gene expression as the processed dsRNAs, deposited in the egg, likely persist and remain active in the embryos. Therefore, wherever possible we have confirmed the phenotypes using just zygotic knockdown (see below) wherein the Gal4 protein and UAS RNAi transgenes are provided by the mother and the father, respectively. Furthermore, although it is thought that *zld* is not required during oogenesis, it is possible that *zld* has a subtle and as yet unknown role in the assembly and/or activity of the maternal PGC determinants. Thus, we tested whether a strictly zygotic knockdown of *zld* also impacted the number of PGCs. In this case, we mated *mat-Gal4* (i.e. *67.15*) virgin females to males carrying *UAS-zldi* RNAi (*zldi*[2z]) or *UAS-egfpi* as a control (*egfpi*[z]) (*Colonnetta et al., 2021a*; *Duan et al., 2021*). While the *egfpi*[z] embryos had an average of 19.7 PGCs (n=10), the *zldi*[2z] embryos had only 12.9 PGCs on average (n=8; p=0.002 by t-test) (*Table 1*). This finding indicates that PGC formation is impacted by reduced *zld* activity in the embryo, and not due to some unexpected effect during oogenesis.

We wondered whether the effects of compromising *zld* on PGC formation are specific to this ZGA factor, or whether other factors implicated in ZGA might have similar phenotypes. To address this question, we used a similar zygotic knockdown strategy to reduce the activity of CLAMP in early

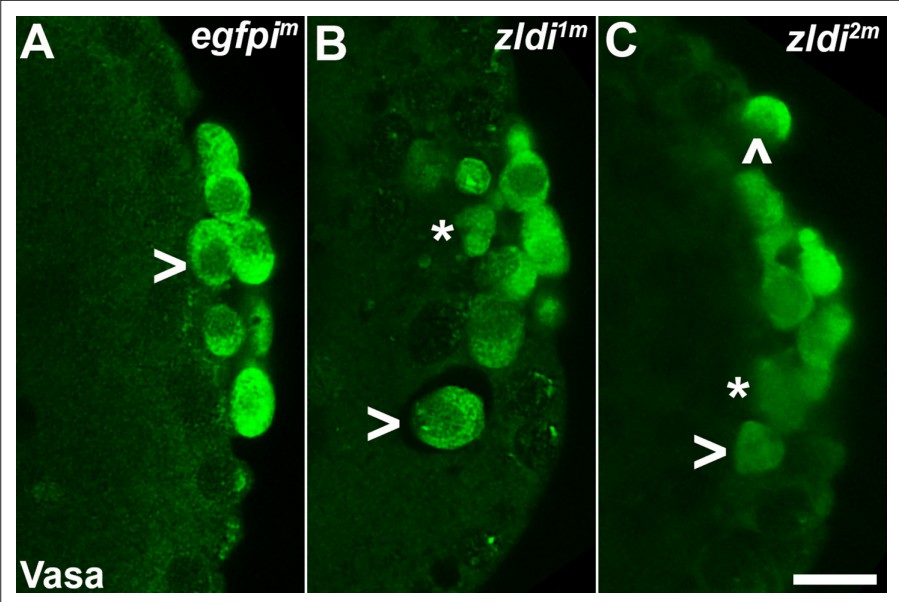

**Figure 1.** *zld* knockdown embryos display variable Vasa levels and abnormal primordial germ cell (PGC) behavior. Zero- to four-hr-old paraformaldehyde-fixed embryos were stained with anti-Vasa antibodies (green) to assess PGC integrity. Shown are representative embryos at nuclear cycle (NC)13 (late syncytial blastoderm stage) of respective genotypes. (**A**) *egfpi^m* embryos have PGCs with uniformly high levels of Vasa (arrowhead, green), but (**B**) *zldi^1m* and (**C**) *zldi^2m* embryos have low levels of Vasa (panel C, asterisk and arrowhead) in PGCs and these PGCs often spread away from the posterior pole in the interior of the embryo (panel B, two such PGCs are marked with an asterisk and an arrowhead). Scale bar represents 10 μm. Images shown are maximum intensity projections through an embryo to capture all PGCs through the Z plane.

The online version of this article includes the following figure supplement(s) for figure 1:

**Figure supplement 1.** *zld* knockdown embryos display a variety of defects in primordial germ cell (PGC) formation.

---

embryos. Like *zldi^2z*, embryos zygotically compromised for CLAMP (*clampi^z*) also have fewer PGCs (12.6; p=0.002 compared to *egfpi^z* control by t-test) (*Table 1*). Thus, diminishing activities of two different ZGA regulators results in a modest but comparable reduction in total PGC number.

## Germ plasm components are mislocalized in embryos compromised for ZGA regulators

Reduction in PGC numbers can be a consequence of a failure to properly incorporate maternal germ-line determinants during PGC cellularization (*Lerit et al., 2017*). To investigate if PGC loss observed upon compromising *zld* or *clamp* function is due to this possibility, we first examined Vasa protein levels in newly formed PGCs. We found that PGCs from the *egfpi^m* control embryos have uniformly high levels of Vasa (n=123) (*Figure 1A*). In contrast, we observed a significant reduction in Vasa protein levels using two different UAS *zld* RNAi transgenes, *zldi^1m* and *zldi^2m* (*Figure 1B and C*), to knock down *zld* mRNA during oogenesis. For *zldi^1m* embryos, 38% of PGCs (n=199; p<0.001 by Fisher's exact test) had reduced levels of Vasa, while in *zldi^2m* embryos, reduced levels of Vasa were observed in 52% of the PGCs (n=187, p<0.001 by Fisher's exact test). In addition to a reduction in Vasa protein, we found that PGCs are located at unusual positions in the *zld* knockdown embryos. In the example shown in *Figure 1B*, there are several PGCs in the interior of the embryo instead of at the surface (marked with an asterisk and an arrowhead in panel B), while there are overlying somatic nuclei. In *Figure 1C*, there is a single PGC (arrowhead) located on the dorsal side of the embryo away from the posterior PGC cluster while the rest of the PGCs are clustered together albeit most show reduced level of Vasa which are marked (see legend).

Like Vasa protein, we found that mRNAs encoding the germline determinants *pgc* and *gcl* are mislocalized when *zld* function is compromised. *pgc* mRNA is not tightly sequestered in PGCs located

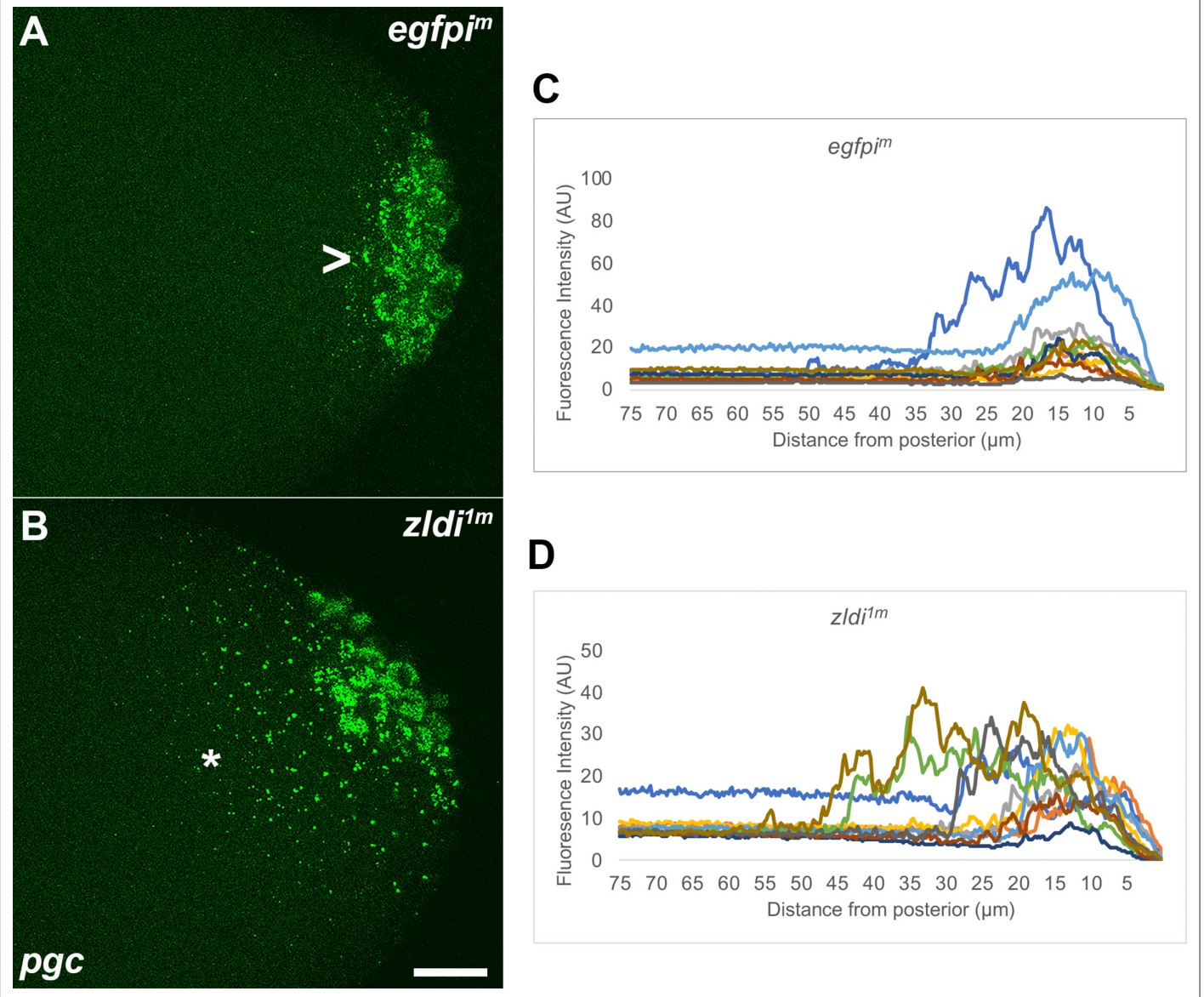

**Figure 2.** Ectopic localization of *polar granule component (pgc)* RNA away from the posterior pole in syncytial and cellular blastoderm *zld* knockdown embryos. Single molecule fluorescence in situ hybridization (smFISH) was performed using probes specific for *pgc* (green) on 0- to 4-hr-old paraformaldehyde-fixed (**A**) *egfpi^m* and (**B**) *zldi^1m* embryos to assess germ plasm localization. Scale bar represents 10 µm. Images shown are representative maximum intensity projections through nuclear cycle (NC)13 (late syncytial blastoderm) embryos to capture total germ plasm localization in Z. Sequestration to the posterior cap is shown with an arrowhead while germ plasm spread into the soma is indicated with an asterisk. The degree of germ plasm spread was measured by fluorescence intensity from maximum projections (visualized using *pgc*) in the posterior 75 µm of (**C**) *egfpi^m* and (**D**) *zldi^1m* embryos. Each plot shows a representative experiment, with each line depicting germ plasm distribution of an individual embryo (see Materials and methods for details of quantification).

at posterior pole in *zldi^1m* embryos, but instead spreads anteriorly so that the mRNA is associated with nearby somatic nuclei (**Figure 2**). The distribution profile of *pgc* mRNA in several *egfpi^m* and *zldi^1m* embryos is shown in **Figure 2C and D**. The spreading of germ plasm mRNAs is not restricted to *pgc* as other germ plasm mRNAS such as *gcl* (**Figure 3**) and *osk* (**Figure 3—figure supplement 1**) are also not properly sequestered in cellularizing PGCs in *zldi^1m* knockdown embryos. We found that germ plasm mRNAs spread away from the posterior in 57% of *zldi^1m* (n=110; p<0.001) embryos while spreading was observed in only 12.1% of control *egfpi^m* embryos (n=99) (e.g. dark blue trace in panel C). Similar results were obtained for *zldi^2m* embryos: spreading was observed in 57% of *zldi^2m* (n=23; p<0.001).

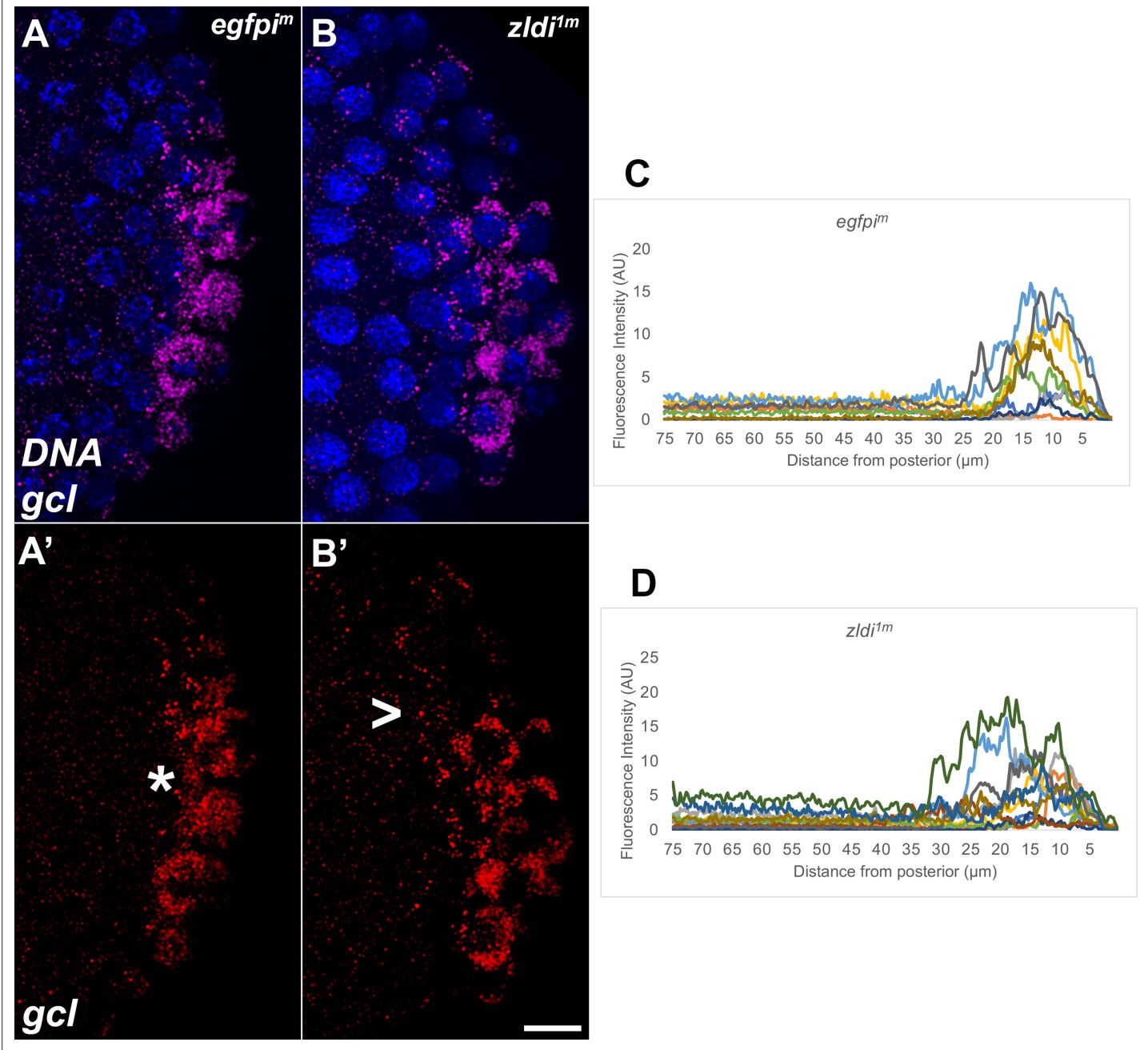

**Figure 3.** Syncytial and cellular blastoderm *zld* knockdown embryos display partially mislocalized *gcl* RNA. Single molecule fluorescence in situ hybridization (smFISH) was performed using probes specific for *gcl* (red) on 0- to 4-hr-old paraformaldehyde-fixed (**A,A'**) *egfpi*ᵐ (nuclear cycle [NC]12) and (**B,B'**) *zldi*¹ᵐ (NC12) embryos to assess *gcl* RNA localization. Scale bar represents 10 μm. Images shown are representative maximum intensity projections through NC12 (late syncytial blastoderm) embryos to capture total germ plasm localization in Z. Sequestration to the posterior cap is shown with an asterisk while germ plasm spread into the soma is indicated with an arrowhead. The degree of germ plasm spread was measured by fluorescence intensity from maximum projections (visualized using *gcl*-specific signal) in the posterior 75 μm of (**C**) *egfpi*ᵐ and (**D**) *zldi*¹ᵐ embryos. Each plot shows a representative experiment, with each line depicting germ plasm distribution of an individual embryo (see Materials and methods for details of quantification).

The online version of this article includes the following figure supplement(s) for figure 3:

**Figure supplement 1.** Ectopic localization of *osk* RNA in syncytial blastoderm *zld* knockdown embryos.

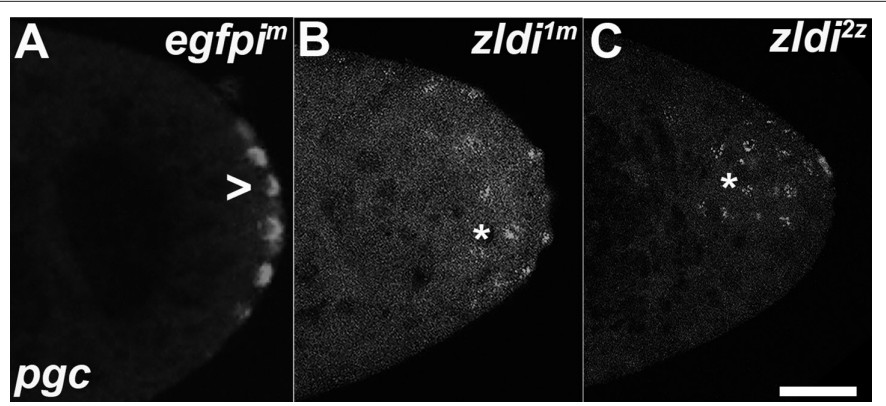

**Figure 4.** Zygotic knockdown of *zld* recapitulates the ectopic localization of *polar granule component* (*pgc*) transcripts induced upon its maternal loss. Single molecule fluorescence in situ hybridization (smFISH) was performed using probes specific for *pgc* on 0- to 3-hr-old paraformaldehyde-fixed (**A**) *egfpi^m*, (**B**) *zldi^1m*, and (**C**) *zldi^2z* embryos to assess germ plasm localization. Scale bar represents 10 µm. *pgc* RNA is seen at the posterior cortex in an nuclear cycle (NC)11 (early syncytial blastoderm) *egfpi^m* embryo (marked with an arrowhead) whereas both *zldi^1m* and *zldi^2z* embryos at NC10/11 (early syncytial blastoderm) show similar mislocalization of *pgc* RNA (marked with an asterisk in both panels).

The online version of this article includes the following figure supplement(s) for figure 4:

**Figure supplement 1.** Zygotic knockdown of *CLAMP* recapitulates the germ plasm mislocalization phenotype.

**Figure supplement 2.** *zld* and *clamp* knockdown embryos display variable spreading of germ plasm RNAs.

As in the case of Vasa protein, ectopic localization of the germ plasm in syncytial blastoderm stage embryos does not appear to be due simply to some unknown defect in germ plasm assembly caused by the depletion of *zld* mRNA during oogenesis. However, maternal RNAi-based knockdown can also influence zygotic gene expression. Thus, we sought to examine whether just zygotic knockdown yields similar phenotypic consequences. We found that germ plasm mRNAs are not properly sequestered in cellularizing PGCs even when *zld* is knocked down zygotically. As illustrated in *Figure 4*, *pgc* mRNAs are found associated with somatic nuclei that are located away from the posterior pole in early syncytial blastoderm stage *zldi^1m* (*Figure 4B*) and *zldi^2z* (*Figure 4C*) embryos unlike in control *egfpi^z* embryos (*Figure 4A*). Moreover, this is not unique to *zld* knockdowns, as spreading of germ plasm mRNAs into the soma is also observed in *clampi^z* zygotic knockdowns. *Figure 4—figure supplement 1* shows that *gcl* mRNAs associate with somatic nuclei that are distant from posterior pole in *clampi^z* knockdown embryos, while *gcl* mRNAs are properly localized in the *egfpi^z* control (compare *Figure 4—figure supplement 1A and B*). As the zygotic knockdown phenotypes appear to qualitatively resemble the maternal knockdown, together these data suggest that Zld and CLAMP contribute to germline/soma distinction zygotically.

To better understand the relationship between the progression of the nuclear division cycles and germ plasm spread in *zldi1^m* embryos, we examined maximum intensity projections of embryos labeled with *pgc* and *gcl* and/or *osk* probes. We classified the labeled embryos using two independent criteria: degree of germ plasm spread and embryonic stage (judged by the NC determined by nuclear density) (*Table 2*). Germ plasm localization defects were denoted as 'none' (all germ plasm constituents effectively sequestered to pole buds/PGCs), 'moderate' (germ plasm RNAs are ectopically localized but are seen associated with the somatic nuclei neighboring PGCs), and 'severe' (germ plasm components are detected well beyond immediately adjacent somatic nuclei). Based on their respective age, embryos were also classified into two categories: early syncytial blastoderm (ESB: NC10 and -11), or late syncytial blastoderm (LSB: NC12 and -13). As can be seen in *Table 2*, germ plasm mRNAs that were not captured in the PGCs during their cellularization continue to spread into the surrounding soma as the NCs proceed. Spreading after PGC cellularization could be due simply to the diffusion of mRNAs that were not captured in PGCs; alternatively, since the mRNAs often accumulate in clusters surrounding the somatic nuclei (asterisk in *Figure 3—figure supplement 1*), it is

**Table 2.** Mislocalization of germ plasm increases in *zld*-compromised embryos as embryogenesis progresses.

Embryos were staged as either early syncytial blastoderm (ESB, NC10-11) or late syncytial blastoderm (LSB, NC12-13), and degree of germ plasm spread defect was classified based on localization of *pgc* and/or *gcl* RNAs.

| Genotype | Stage | None | Moderate | Severe | Total |
|---|---|---|---|---|---|
| | | | **Defects** | | |
| *egfpi*[m] | | **66 (84.6%)** | **12 (15.4%)** | **0 (0%)** | **78** |
| | ESB | 24 (85.7%) | 4 (14.3%) | 0 (0%) | 28 |
| | LSB | 42 (84%) | 8 (16%) | 0 (0%) | 50 |
| *zldi*[1m] | | **24 (27.9%)** | **49 (57%)** | **13 (15.1%)** | **86** |
| | ESB | 11 (34.3%) | 15 (46.9%) | 6 (18.8%) | 32 |
| | LSB | 13 (24%) | 34 (63%) | 7 (13%) | 54 |

The online version of this article includes the following source data for table 2:

**Source data 1.** Raw data summarized in *Table 2*: Embryonic stage and pole plasm spreading phenotype for individual embryos.

possible that a microtubule-dependent mechanism actively transmits germ plasm RNAs through the soma.

## PGCs from embryos compromised for Zld or CLAMP display aberrant centrosome behavior

The initial formation of PGCs depends upon at least one germ plasm component, *gcl*, and the proper functioning of the centrosomes associated with the PGC nuclei (*Jongens et al., 1992*; *Lerit et al., 2017*; *Raff and Glover, 1989*; *Robertson et al., 1999*). When they enter the posterior pole, centrosomes trigger the release of the germline determinants from the cortex and subsequently direct the trafficking and partitioning of the germ plasm into the cellularizing PGCs. When centrosome function is disrupted, germ plasm is not properly incorporated into the PGCs and instead spreads into the surrounding soma (*Lerit et al., 2017*; *Lerit and Gavis, 2011*). In their initial characterization of *zelda*, *Staudt et al., 2006*, showed that *zelda* mutant embryos had a range of nuclear division defects including asynchronous replication and improper chromosome segregation. Consistent with their observation, we found that centrosome defects are observed in syncytial blastoderm embryos zygotically compromised for either *zld* or *clamp* (*Colonnetta et al., 2021a*). We thus sought to determine if embryos maternally compromised for Zld show similar defects and therefore analyzed embryos by immunostaining for Centrosomin (Cnn), a core component of mitotic centrosomes, and Peanut (Pnut), a *Drosophila* Septin, to visualize the cytoskeleton. Confirming a requirement for maternally deposited *zelda*, we observed comparable centrosome defects in the *zldi*[1m] embryos. In the late syncytial blastoderm stage embryos (NC13-14), nearly all (15/18; 88%) of the *zldi*[1m] embryos had centrosome defects as opposed to (2/15; 11%) control *egfpi*[m] embryos (p<0.001 by Fisher's exact test). As illustrated in *Figure 5* (arrowhead), a common defect was a lack of proper separation after centrosome duplication. In wild type (WT), centrosomes duplicate early in the nuclear division cycle and then immediately move to opposite poles of each nucleus. For this reason, duplicated but not yet separated centrosomes are rarely observed in WT embryos. Other defects include 'orphan' centrosomes that are not associated with nuclei, and multiple (fragmented, asterisk) centrosomes associated with a single nucleus.

Next, we wondered whether the centrosome defects seen at the late syncytial blastoderm stage (NC13-14) can be traced back to earlier NCs during development. We thus stained 0- to 3-hr-old embryos maternally compromised for *zelda*, which contained both NC10-11 and even younger pre-syncytial embryos between NC1-10. As shown in *Figure 6*, we observed aberrant centrosome separation in the somatic nuclei from cycle 11 embryos.

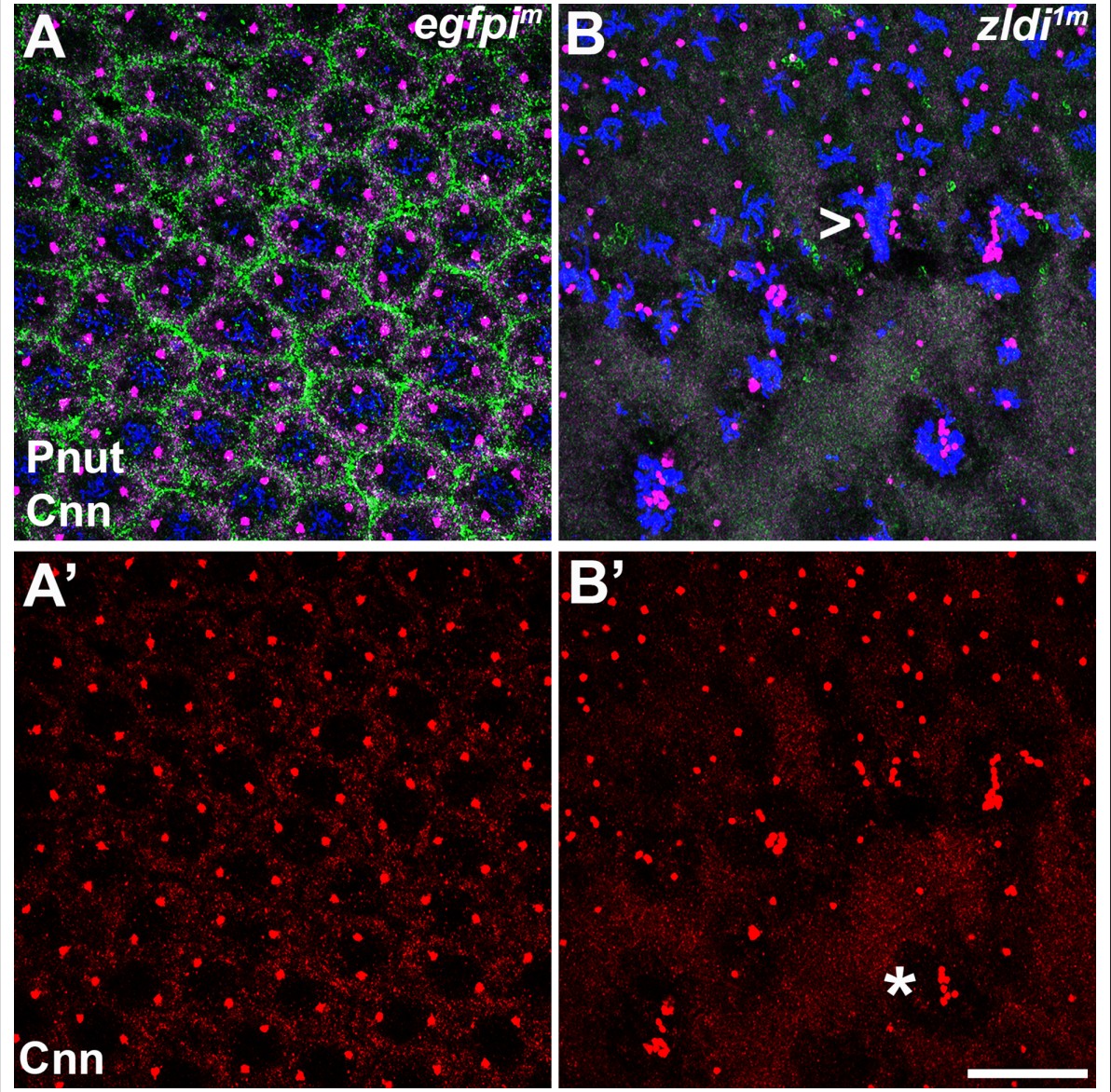

**Figure 5.** Centrosomes display numerous aberrations in soma of *zld*-compromised embryos. Zero- to four-hr old paraformaldehyde-fixed embryos were stained with anti-Centrosomin (anti-Cnn) and anti-Peanut (anti-Pnut) antibodies to assess centrosome behavior in the soma of *zldi¹ᵐ* embryos. (**A**) *egfpiᵐ* embryo at nuclear cycle (NC)12 (late syncytial blastoderm) has regularly spaced nuclei with two centrosomes (A', red) at opposite poles surrounded by a ring of Pnut (green). (**B**) *zldi¹ᵐ* embryo at a comparable stage displays numerous defects, including duplicated but not separated centrosomes (arrowhead, B', red) and disrupted accumulation of the cytoskeletal protein Pnut. Embryos were co-stained with Hoescht to visualize nuclei (blue). Scale bar represents 10 μm.

Interestingly, duplicated but not separated centrosomes were also seen in even younger, that is, pre-syncytial embryos. Thirty-five percent of the nuclei (n=37) in *zldi¹ᵐ* pre-syncytial embryos (NC7-9) showed this phenotype while only 6% of the nuclei (n=51) in *egfpiᵐ* NC7-9 control embryos had similar centrosome problems (p<0.001 by Fisher's exact test). Thus, centrosome defects are observed even in very young embryos where nuclei have not yet reached the periphery. In addition, in half of the *zldi¹ᵐ* NC7-9 embryos, we also observed at least one fused, oversized nucleus.

The occurrence of centrosome defects when Zld or CLAMP activities are compromised during the pre-syncytial and syncytial nuclear division cycles prompted us to examine centrosome organization during pole bud/PGC formation. In WT pole buds and PGCs, centrosomes duplicate and migrate to opposite pole (27/31; 87% normal while 4/31; 13% showed defects). However, as is the case for the

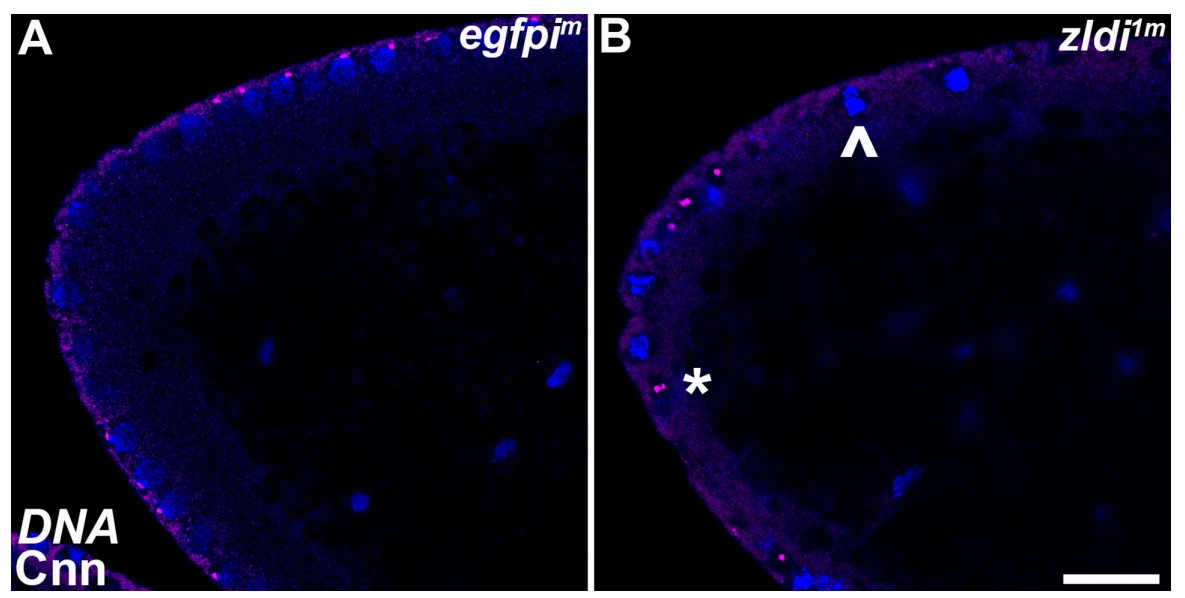

**Figure 6.** Centrosome aberrations are observed in early syncytial blastoderm embryos maternally compromised for *zld*. Zero- to 3-hr-old paraformaldehyde-fixed embryos were stained with anti-Centrosomin (anti-Cnn) (magenta) to assess centrosome behavior in *zldi¹ᵐ* embryos. Embryos were co-stained with Hoechst to visualize nuclei (blue). Both panels show the anterior terminus of a nuclear cycle (NC)11 (early syncytial blastoderm) embryo. (**A**) *egfpiᵐ* embryos have regularly spaced nuclei with correctly segregated centrosomes. (**B**) *zld¹ᵐ* embryos display ineffectively separated centrosomes (asterisk) and fused nuclei (arrowhead). Scale bar represents 10 μm.

somatic nuclei, we observed defective separation of duplicated centrosomes in pole buds and PGCs in the maternal *zld* knockdown, *zldi¹ᵐ* (16/30: 53%, p=0.001 by Fisher's exact test) (*Figure 7*, arrowhead) and in zygotic knockdowns of *zld* (*zldi²ᶻ*) (12/27; 45%, p=0.009 by Fisher's exact test) and *clamp* (*clampiᶻ*) (7/20; 35%, p=0.085 by Fisher's exact test) (*Figure 8*). We also observed 'orphan' centrosomes dissociated from DNA (asterisk) or significantly diffused staining with anti-Cnn antibodies.

## Anterior protrusions in Zld- or CLAMP-compromised embryos

Defects in the process of pole bud formation and PGC cellularization are not the only abnormalities evident in *zld* or *clamp* knockdown embryos in the period just after nuclei migrate to the cortex. At the same time that pole buds are forming at the embryonic posterior and then undergoing cellularization, we observe bud-like protrusions developing at the anterior end of the knockdown embryos. Panels C and D in *Figure 9* show buds forming simultaneously in an early syncytial blastoderm maternal *zldi¹ᵐ* knockdown embryo. Anterior buds like those shown in this figure are observed in about 61% (11/18) of early syncytial blastoderm *zldi¹ᵐ* knockdown embryos, while they are rarely, if ever, observed in maternal *egfpiᵐ* knockdown embryos (9%: n=1/12) (p=0.007 by Fisher's exact test). While we also observe ectopic buds on the dorsal-lateral surface of early syncytial *zldi¹ᵐ* knockdown embryos, these are only seen infrequently, suggesting that the curvature at the anterior might be more permissive for bud formation. As observed for other *zldi¹ᵐ* knockdown phenotypes, ectopic buds are also observed in the zygotic knockdown embryos, though somewhat less frequently, therefore *zldi²ᶻ* (41%; 5/12, p=0.055 by Fisher's exact test) and *clampiᶻ* (30%; 4/13, p=0.131 by Fisher's exact test) knockdowns trend toward significant incidence of protrusions compared to *egfpiᶻ* (0%, 0/8) (*Figure 10*).

## Transcriptional quiescence is compromised in PGCs of ZGA-compromised embryos

Transcriptional quiescence is an important trait that distinguishes PGCs from the surrounding soma. In *D. melanogaster*, maternally deposited germ plasm constituents (*nos*, *gcl*, *pgc*) promote silencing by inhibiting the transcription of overlapping yet distinct sets of target genes. When one of these factors is mutant, a specific set of target genes are transcribed in the mutant PGCs. Since these determinants are not fully captured by the PGCs when they cellularize, we wondered whether this impacted the

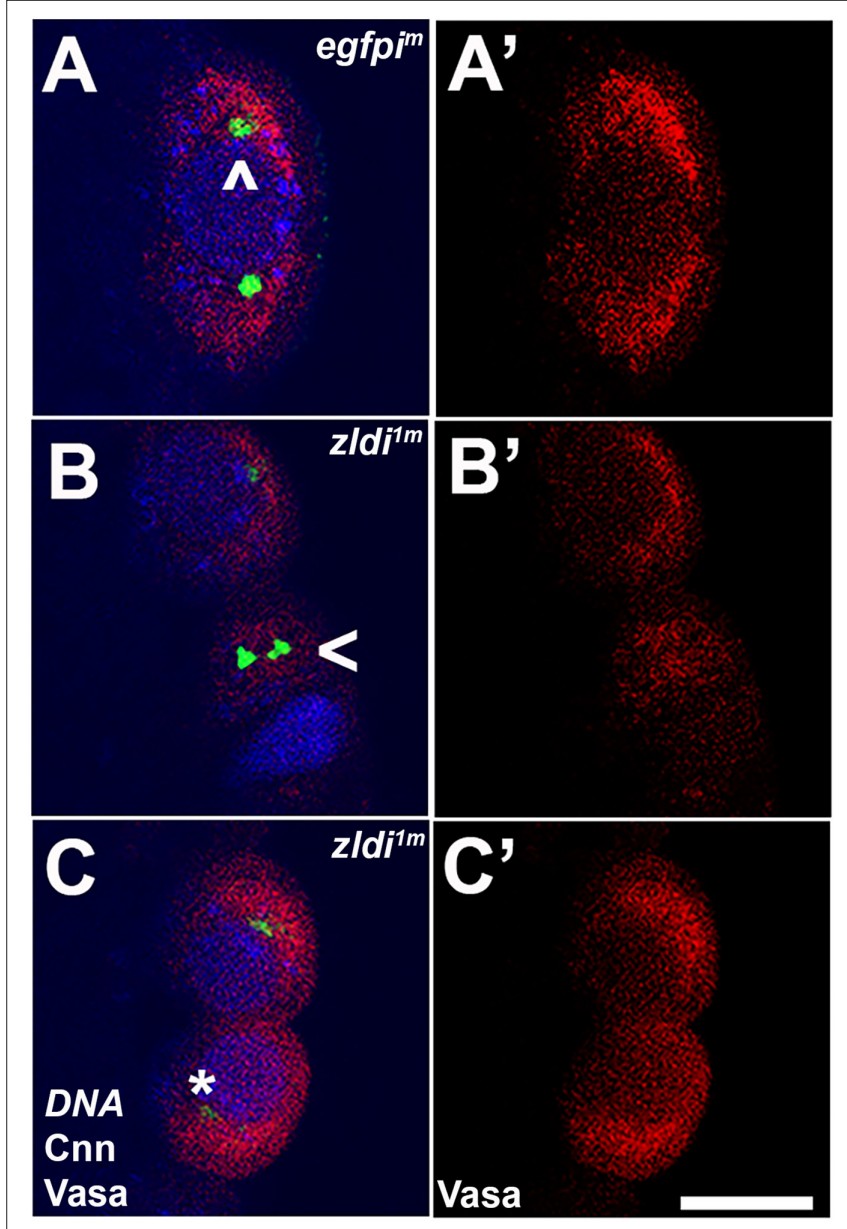

**Figure 7.** *zld*-compromised embryos have defects in centrosome duplication and separation in newly formed primordial germ cells (PGCs). Zero- to three-hr-old paraformaldehyde-fixed embryos were stained with anti-Centrosomin (anti-Cnn) and anti-Vasa antibodies to assess centrosome behavior in the pole buds of *zldi^1m* embryos. (**A**) *egfpi^m* embryos have high levels of Vasa (**A'**, red) in PGCs with centrosomes marked by Cnn (green) at opposite poles of each cell (**A**, caret). (**B**) *zld^1m* embryos display non-segregated centrosomes (**B**, arrowhead), variable levels of Vasa in PGCs (**B'**, red) and reduced level of Cnn (**C**, asterisk, green). Embryos at NC10-11 (early syncytial blastoderm) were co-stained with Hoescht to visualize nuclei (blue). Scale bar represents 10 µm.

establishment of transcriptional quiescence. To test this, we examined the expression of two genes, *Sxl-Pe* and *slam*, which are known to be Zld targets during ZGA (*Liang et al., 2008*; *Nien et al., 2011*; *ten Bosch et al., 2006*). In PGCs, *Sxl-Pe* is kept off by *gcl* and *nos*, while *pgc* blocks the expression of *slam*. In WT embryos, these two genes are never active in PGCs, and this is also true for the *egfpi^m* control embryos (n=30). However, we found that 13.3% of *zldi^1m* embryos (n=15; p=0.106 by Fisher's exact test, not statistically significant) and 7.1% of *zldi^2m* embryos (n=14; p=0.318 by Fisher's exact test, not statistically significant) had *slam* mRNAs in their PGCs (*Figure 11*) while 22.6% of *zldi^1m* embryos (n=31; p=0.011 by Fisher's exact test) and 0% of *zldi^2m* embryos (n=14, p=1.0 by Fisher's

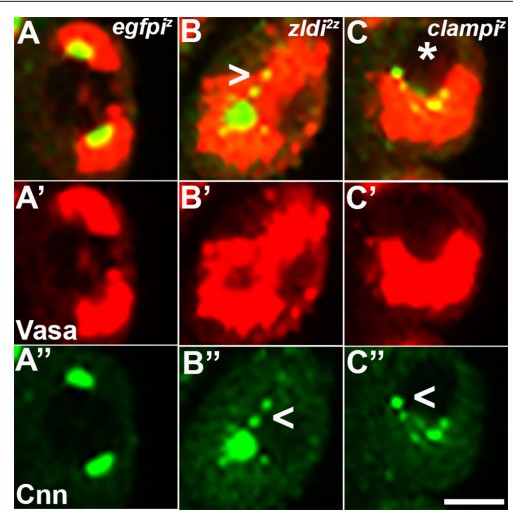

**Figure 8.** Embryos compromised for either *zld* or *clamp* display improperly duplicated and separated centrosomes in newly formed primordial germ cells (PGCs). Zero- to 2.5-hr-old paraformaldehyde-fixed embryos were stained with anti-Centrosomin (anti-Cnn) and anti-Vasa antibodies to assess centrosome behavior in the pole buds of *zld* or *clamp* zygotic knockdown embryos. (**A**) *egfpi^z* embryo at nuclear cycle (NC)10 (early syncytial blastoderm) displays high levels of Vasa (**A'**, red) in PGCs with centrosomes at opposite poles of each cell (**A''**, green). Both *zld^{iz}* (**B**) and *clampi^z* (**C**) embryos also at NC10 (early syncytial blastoderm) exhibit defects including duplicated but attached or improperly separated centrosomes (**B''/C''**, arrowheads) as well as an aberrant pattern of anti-Cnn staining (green). Vasa seems to be unevenly distributed between daughter cells (red, **B'** and **C'**). Scale bar represents 10 μm.

exact test, statistically indistinguishable from control) had nascent *Sxl-Pe* transcripts in their PGCs (*Figure 11—figure supplement 1*). These findings indicate that *zld* is required for properly attenuating transcription in newly formed PGCs.

## Ectopic localization of germ plasm affects somatic transcription

In PGCs, the germline determinants *pgc*, *gcl,* and *nos* function to inhibit RNA PolII activity. When these germline determinants are overexpressed or ectopically expressed, they can downregulate transcription in the soma, interfering with somatic development (*de Las Heras et al., 2009*; *Deshpande et al., 2005*; *Robertson et al., 1999*). Since one of the key roles of germ plasm constituents is establishing transcriptional quiescence in the PGCs, we wondered if somatic transcription is downregulated when germ plasm components spread into surrounding posterior soma. We examined two genes, *tailless* (*tll*) and *decapentaplegic* (*dpp*), that are normally expressed at high levels in both the posterior and anterior of the embryo. The *tll* gene is activated by the terminal pathway, and its activity in PGCs is known to be repressed by *pgc*, which encodes an inhibitor of the PolII CTD elongation kinase pTFB (*Deshpande et al., 2004*; *Hanyu-Nakamura et al., 2008*; *Martinho et al., 2004*), and likely also *gcl*, which promotes the degradation of the terminal pathway receptor *torso* (*Colonnetta et al., 2021b*; *Pae et al., 2017*).

For this reason, the spreading of these germ plasm components into the surrounding soma when ZGA regulators like *zld* are knocked down could potentially suppress *tll* transcription in the posterior. Since *tll* is also a transcriptional target of *zld,* its expression is expected to be downregulated at both the anterior and posterior ends of the embryo when *zld* activity is compromised. This is indeed the case (8/11 *zldi^{1m}* embryos showed discernible reduction in *tll* transcript levels as opposed to 0/10 *egfpi^m* embryos, p=0.001 by Fisher's exact test). However, as illustrated in *Figure 12*, the effects of *zld* knockdown are more severe in the posterior in the same region of the embryo where *gcl* mRNAs spread into the soma. In control embryos, *tll* is expressed at both anterior and posterior termini in the soma of syncytial blastoderm embryos (*Figure 12*). Quantification of the number of nuclei that exhibit *tll*-specific signal indicates that the reduction in *tll* expression at the posterior of the *zldi^{1m}* embryos is significantly more severe than at the anterior (292/397; 74% nuclei at the anterior as opposed to 132/403; 33% nuclei in the posterior showed *tll*-specific signal, p<0.001 by Fisher's exact test. See Materials and methods for details).

Next, we investigated transcription of *dpp*, another direct target of Zld (*Liang et al., 2008*; *Nien et al., 2011*) and a key signaling ligand of the BMP family (*Deignan et al., 2016*; *Matsuda et al., 2016*; *O'Connor et al., 2006*). Using exonic probes, we visualized zygotic transcription of *dpp* which is activated by NC10 concomitantly with pole bud formation and PGC cellularization. Subsequently, as global transcription is upregulated during NC12-14 and the dorsal-ventral axis of the embryo is established, the levels of *dpp* transcripts are substantially increased on the dorsal half of the embryo including both the poles (*Chang et al., 2001*; *Stathopoulos et al., 2002*). As shown for *efgpi^m* in *Figure 13*, the expression domain at the posterior terminus of the embryo encompasses the pole

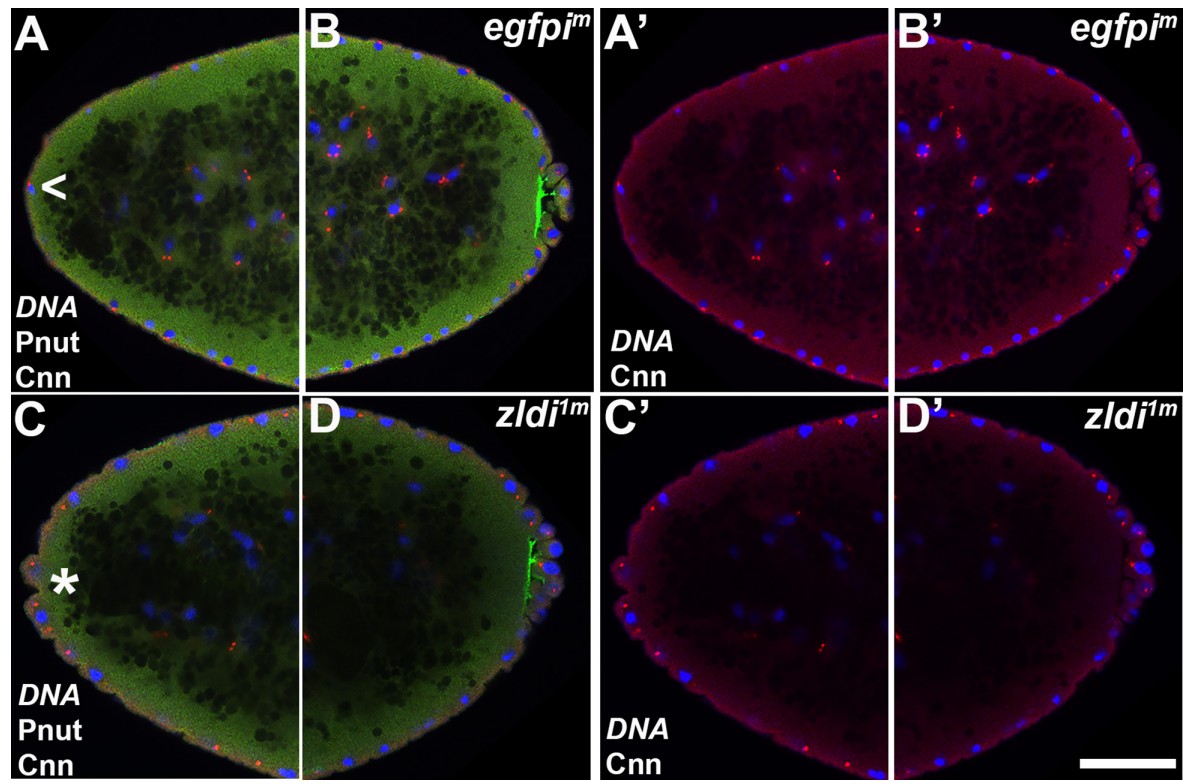

**Figure 9.** Appearance of ectopic protrusions at the anterior pole of maternal *zld* knockdown embryos. Zero- to four-hr-old paraformaldehyde-fixed embryos were stained with anti-Centrosomin (anti-Cnn) and anti-Peanut (anti-Pnut) antibodies to assess morphology at the poles. Anterior (**A,A',C,C'**) and posterior (**B,B'/D,D'**) views of a single representative *egfpi^m* (A,A'/B,B') or *zldi^1m* (C,C'/D,D') embryos at nuclear cycle (NC)12 (late syncytial blastoderm) are shown with nuclei (Hoescht, blue), centrosomes (Cnn, red), and cytoskeleton (Pnut, green). Though nuclei and their associated centrosomes reach the anterior terminus of control embryos, they do not bulge outward (arrowhead, **A**). In some *zldi^1m* embryos, nuclei and their associated centrosomes protrude outward, mirroring pole bud behavior (asterisk, **C**) Scale bar represents 10 μm.

cells. Confirming that *dpp* is a target for *zld*, in 10/13 *zldi^1m* knockdown embryos, *dpp* expression in the dorsal half of the embryo is reduced as compared to 0/11 *efgpi^m* control embryos (p<0.001 by Fisher's exact test; compare *Figure 13*, panels A' and B'). *dpp* expression at the anterior terminus is also reduced compared to the *egfpi^m* control embryos. Importantly, as was observed for *tll*, the reduction in the transcript levels is typically more pronounced in the posterior soma in the regions of the embryo where mislocalized *pgc* mRNA can be detected (*Figure 14*). We quantified the number of nuclei expressing *dpp* versus the total number of nuclei in the anterior and posterior halves of *zldi^1m* embryos (see Materials and methods for details of quantification). In the anterior 77/119; 64% of the nuclei expressed *dpp* while in the posterior only 38/129; 29% of the nuclei expressed *dpp* (p<0.001 by Fisher's exact test).

## Discussion

The conserved process of ZGA is primarily responsible for providing a roadmap that engineers gradual transition of a single cell into a multicellular organism consisting of diverse cell types capable of performing defined functions. To be able to perform distinct functions in a coordinated manner, it is imperative that differentiated cells acquire precise spatiotemporal coordinates and corresponding unique functional attributes. Thus, ZGA transforms a totipotent fertilized zygote into a multicellular assembly made up of various terminally differentiated cell types (*Hamm and Harrison, 2018*; *Schulz and Harrison, 2019*; *Tadros and Lipshitz, 2009*).

In *D. melanogaster*, ZGA consists of a minor and a major wave, which are temporally separable. The proper specification of distinct somatic cell types in the *Drosophila* embryo takes place during the major wave of ZGA. The global activation of transcription is dependent upon at least three

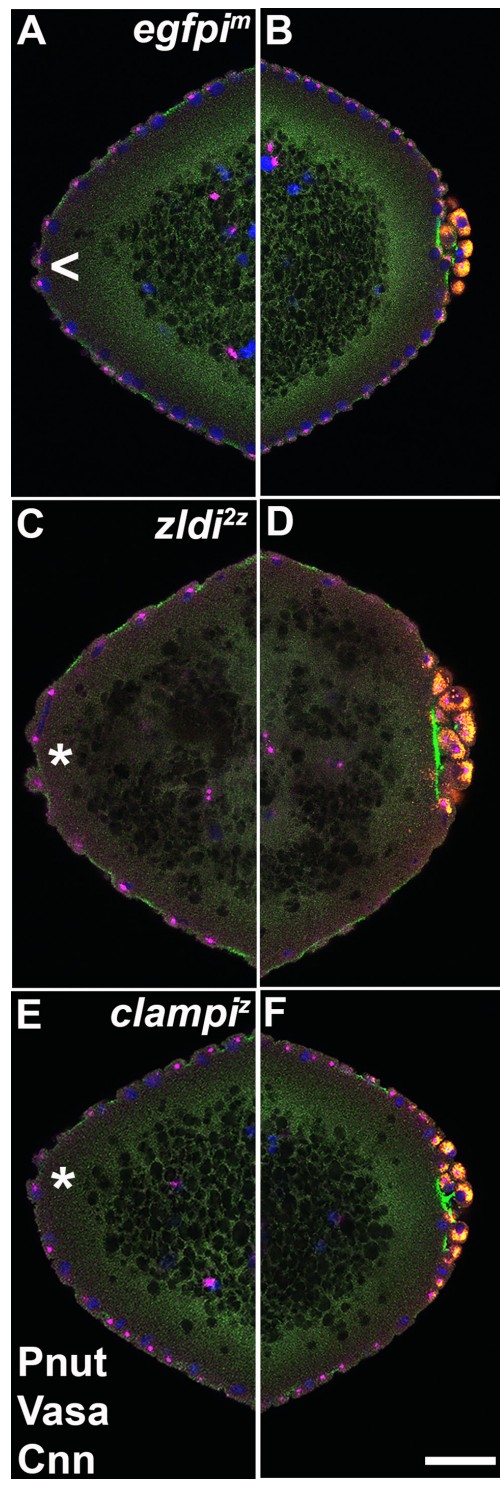

*Figure 10 continued*

poles. Anterior (A/C/E) and posterior (B/D/F) views of a single representative *egfpi^m* (A/B), *zldi^{2z}* (C/D), or *clampi^z* (E/F) embryo are shown with nuclei (Hoescht, blue), centrosomes (Cnn, magenta), cytoskeleton (Pnut, green), and PGCs (Vasa, red). (All three embryos are nuclear cycle [NC]12, late syncytial blastoderm stage.) Though nuclei and their associated centrosomes reach the anterior terminus of control embryos, they do not bulge outward (arrowhead, **A**). In some *zld^{2z}* embryos, nuclei and their associated centrosomes protrude outward, mirroring pole bud behavior (asterisk, **C**). *clampi^z* embryos also form anterior bulges (asterisk, **E**). Scale bar represents 10 μm.

**Figure 10.** At the anterior pole of zygotic *zld* and *clamp* knockdown embryos, protrusions appear shortly after the formation of primordial germ cells (PGCs) at the posterior pole. Zero- to four-hr-old paraformaldehyde-fixed embryos were stained with anti-Peanut (anti-Pnut), anti-Vas, and anti-Centrosomin (anti-Cnn) antibodies to assess morphology at the

*Figure 10 continued on next page*

genome-wide regulators, Zld, CLAMP, and GAF (*Colonnetta et al., 2021a*; *Duan et al., 2021*; *Harrison et al., 2011*; *Liang et al., 2008*; *Nien et al., 2011*). While the most substantial upregulation of transcription during the major ZGA wave occurs in NC14, the minor wave, which commences at NC8, involves only relatively sparse and low levels of transcription. Nonetheless, genes transcribed during this period regulate two crucial processes: somatic sex determination and early patterning. Coincident with these two steps in fate specification, another important developmental decision, namely germline versus somatic identity, also takes place. Here, we have investigated the possible role of the factors that mediate ZGA in distinguishing germline versus soma. Our results indicate that two regulators of ZGA, Zld and CLAMP, play important roles in establishing germline and somatic fate during ZGA.

We find that compromising either *zld* or *clamp* by RNAi knockdown has reciprocal effects on cells normally destined to be either germline (PGC) or soma. In the former case, newly formed PGCs exhibit a range of defects consistent with a partial transformation toward somatic identity. These include a loss or reduction in the levels of the germline specific marker Vasa and the upregulation of transcription of genes (*Sxl-Pe* and *slam*) that are never transcribed in WT PGCs. Conversely, in the soma, we observed reductions in the level of transcription of the *tll* and *dpp* genes at the posterior of embryo over and above that caused by the embryo-wide loss of Zld activity. The observed phenotypes exhibiting the disruption in both germline and somatic fates are summarized in *Table 3*.

The perturbations in both germline and soma specification can be traced back to defects in the process of PGC cellularization. In WT, the centrosomes associated with the nuclei migrating into the posterior pole trigger the release of germline

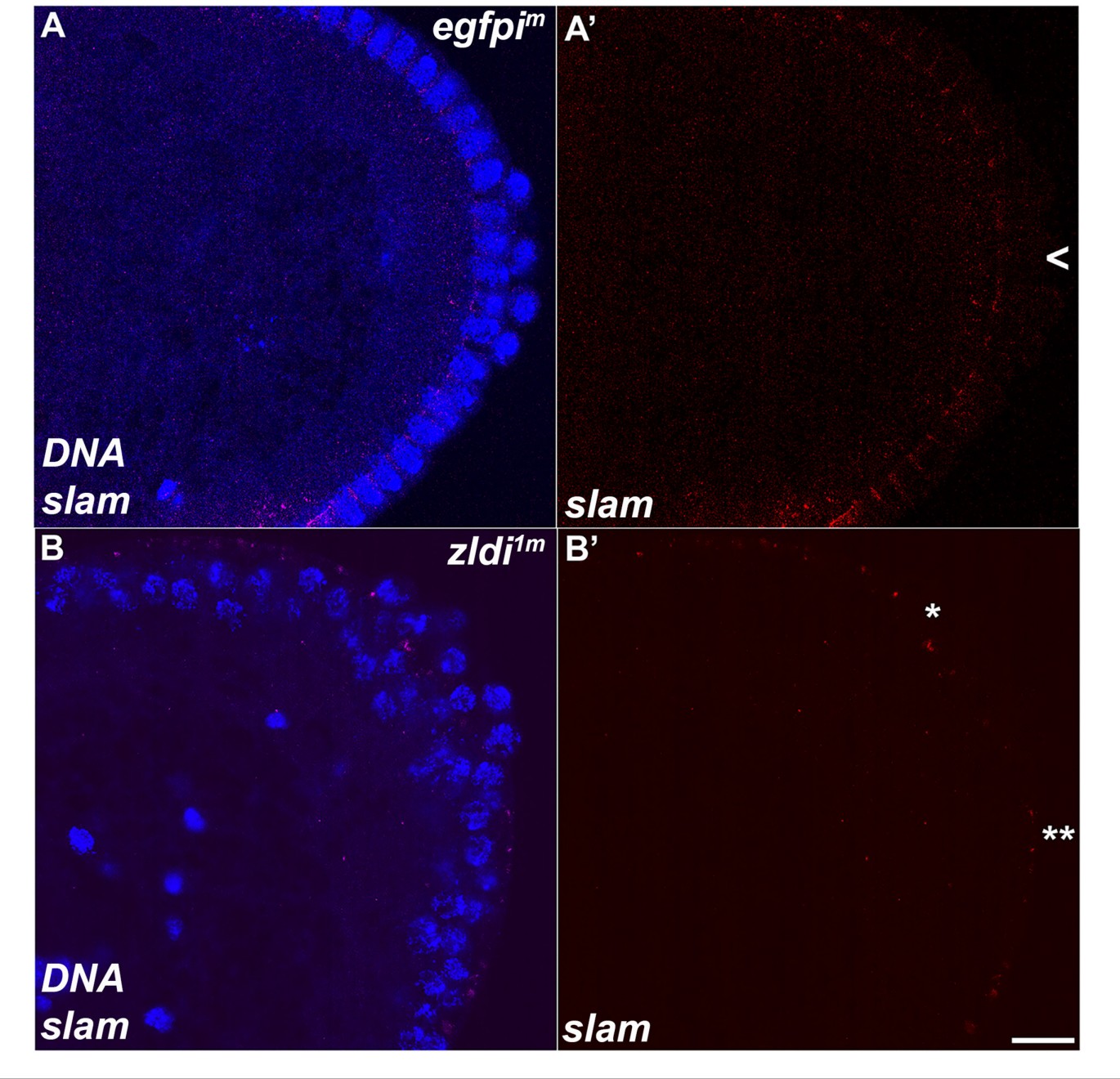

**Figure 11.** Unlike wild type, a subset of embryonic primordial germ cells (PGCs) from *zld*-compromised embryos displays *slam* transcripts. Single molecule fluorescence in situ hybridization (smFISH) was performed using probes specific for *slam* on 0- to 4-hr-old embryos to assess the status of transcription in PGCs. While *slam* is never expressed in (**A**) *egfpi^m* control PGCs, some (**B**) *zldi^1m* PGCs display ectopic *slam* transcription (asterisk). Posterior poles of representative syncytial blastoderm embryos (nuclear cycle [NC]13, late syncytial blastoderm) are shown with *slam* RNA visualized in magenta and Hoescht DNA dye in blue. Scale bar represents 10 µm.

The online version of this article includes the following figure supplement(s) for figure 11:

**Figure supplement 1.** *zld*-compromised embryos ectopically express *Sxl-Pe* in embryonic primordial germ cells (PGCs).

determinants that are anchored to the cortex during oogenesis. After release, the germline determinants are trafficked on astral microtubules so they encompass the pole bud nuclei (*Lerit et al., 2017*; *Lerit and Gavis, 2011*). When cellularization takes place the germline determinants are efficiently captured in the newly formed PGCs. In the maternal *zld* knockdown and in the zygotic *zld* and *clamp* knockdowns, the sequestration of the germline determinants during PGC cellularization is aberrant

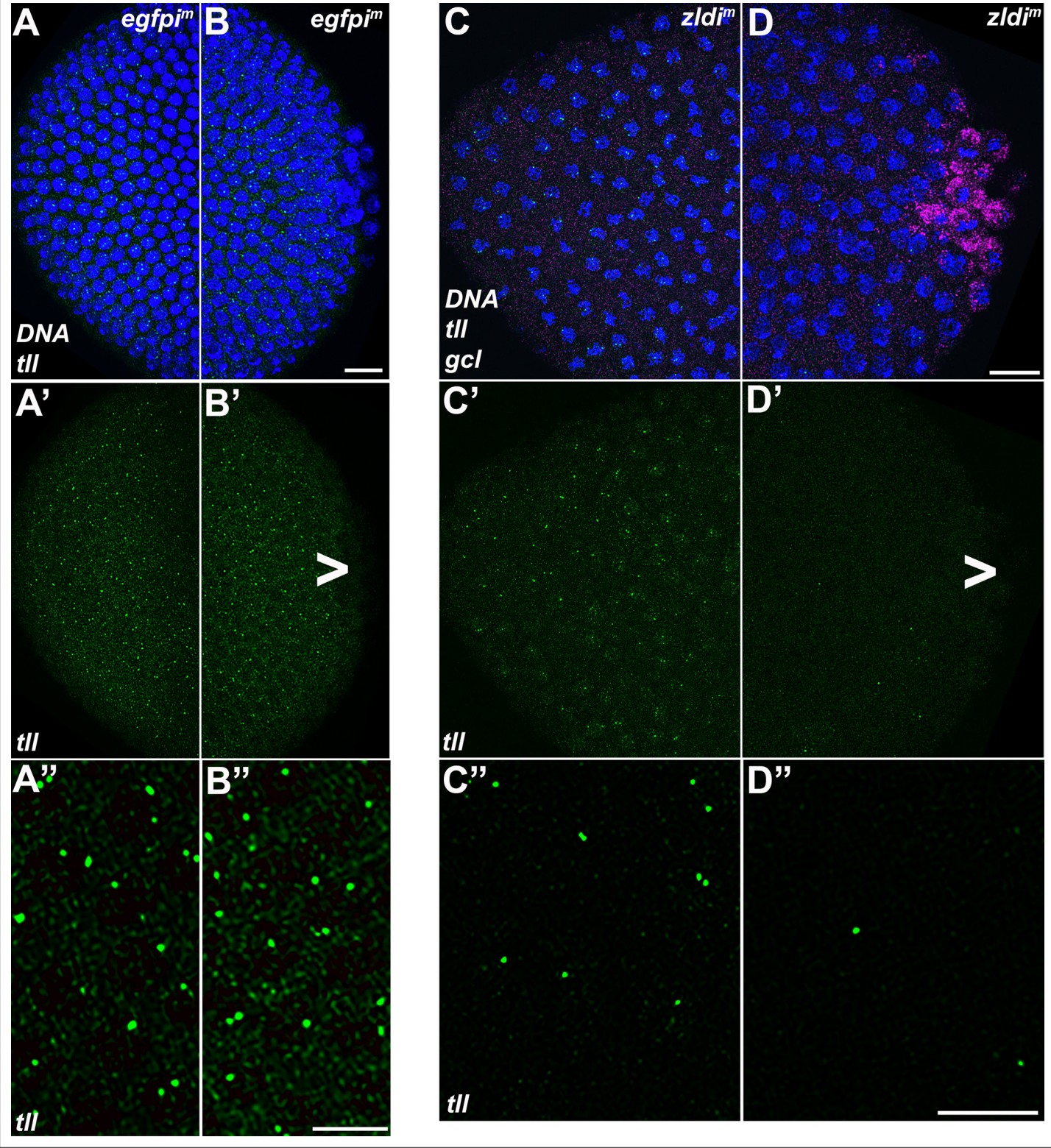

**Figure 12.** *tll* somatic transcription is notably decreased in the embryonic posterior upon *zld* knockdown. Single molecule fluorescence in situ hybridization (smFISH) was performed using probes specific for *tll* (green) and *gcl* (magenta) on 0- to 4-hr-old egfp*i*$^m$ (A/B) and zldi$^{1m}$ (C/D) embryos to assess the status of somatic transcription at the anterior and posterior poles. Maximum intensity projections of anterior (**A–A″,C–C″**) and posterior (**B–B″,D–D″**) poles of representative late syncytial blastoderm (nuclear cycle [NC]13) embryos are shown with *tll* RNA visualized in green, *gcl* RNA in magenta, and Hoescht DNA dye in blue. Scale bar represents 10 μm. While *tll* transcription is comparable between anterior (**A′ and A″**) and posterior

*Figure 12 continued on next page*

*Figure 12 continued*

(**B' and B"**) termini of *egfpi^m* embryos, transcript levels are reduced in *zldi^{1m}* embryos, to a greater extent at the posterior pole (**D' and D"**) as compared to the anterior (**C' and C"**) *tll* is not expressed in primordial germ cells (PGCs) (arrowhead).

and these factors disperse into the nearby soma. The failure to incorporate the full complement of germline determinants is responsible at least in part for the PGC phenotypes observed in the *zld* and *clamp* RNA knockdowns. Consistent with this conclusion, similar PGC phenotypes are observed in *gcl* mutants and in mutants in components of the BMP signaling pathway (*Colonnetta et al., 2021b*; *Colonnetta et al., 2022*). In both of these cases, germline determinants are not efficiently captured by the cellularizing PGCs. Likewise, the ectopic presence of germline determinants in a somatic domain is likely responsible for exacerbating the reductions in transcription of *tll* and *dpp* in the posterior soma of *zldi^{1m}* knockdown embryos. This is supported by earlier observations showing that *pgc*, *gcl*, and *nos* can downregulate transcription in the soma upon overexpression or ectopic expression (*de Las Heras et al., 2009*; *Deshpande et al., 2005*; *Robertson et al., 1999*). Likely there are many other genes that are active in the posterior region of blastoderm stage embryos whose expression is downregulated

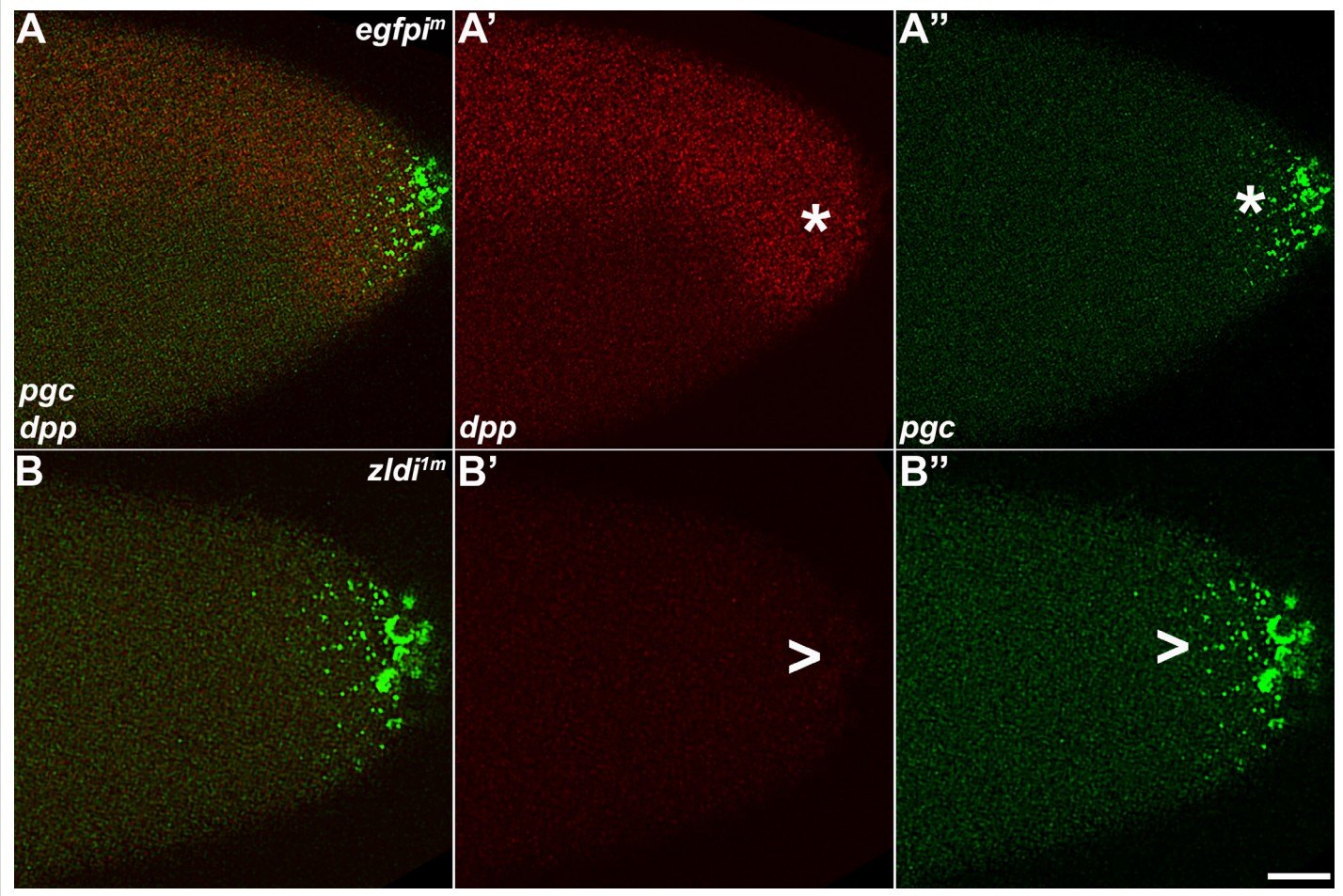

**Figure 13.** As the germ plasm RNAs spread into the posterior soma, somatic transcription of *decapentaplegic (dpp)* decreases in *zld*-compromised embryos. Single molecule fluorescence in situ hybridization (smFISH) was performed using probes specific for *dpp* (**A' and B'**) and *polar granule component (pgc)* (**A" and B"**) on 0- to 4-hr-old embryos to assess the status of somatic transcription at the posterior pole. *egfpi^m* embryos (**A–A"**) express *dpp* dorsally (red, B, asterisk), but these expression levels are compromised in *zldi^{1m}* embryos (**B–B"**,B', arrowhead), particularly those displaying more severe germ plasm spread as shown by *pgc* (green, marked with an arrowhead in panel B"). Scale bar represents 10 µm. Images shown are maximum intensity projections through embryos at nuclear cycle (NC)13 (late syncytial blastoderm) to capture total germ plasm localization and *dpp* expression throughout relevant focal planes (in Z).

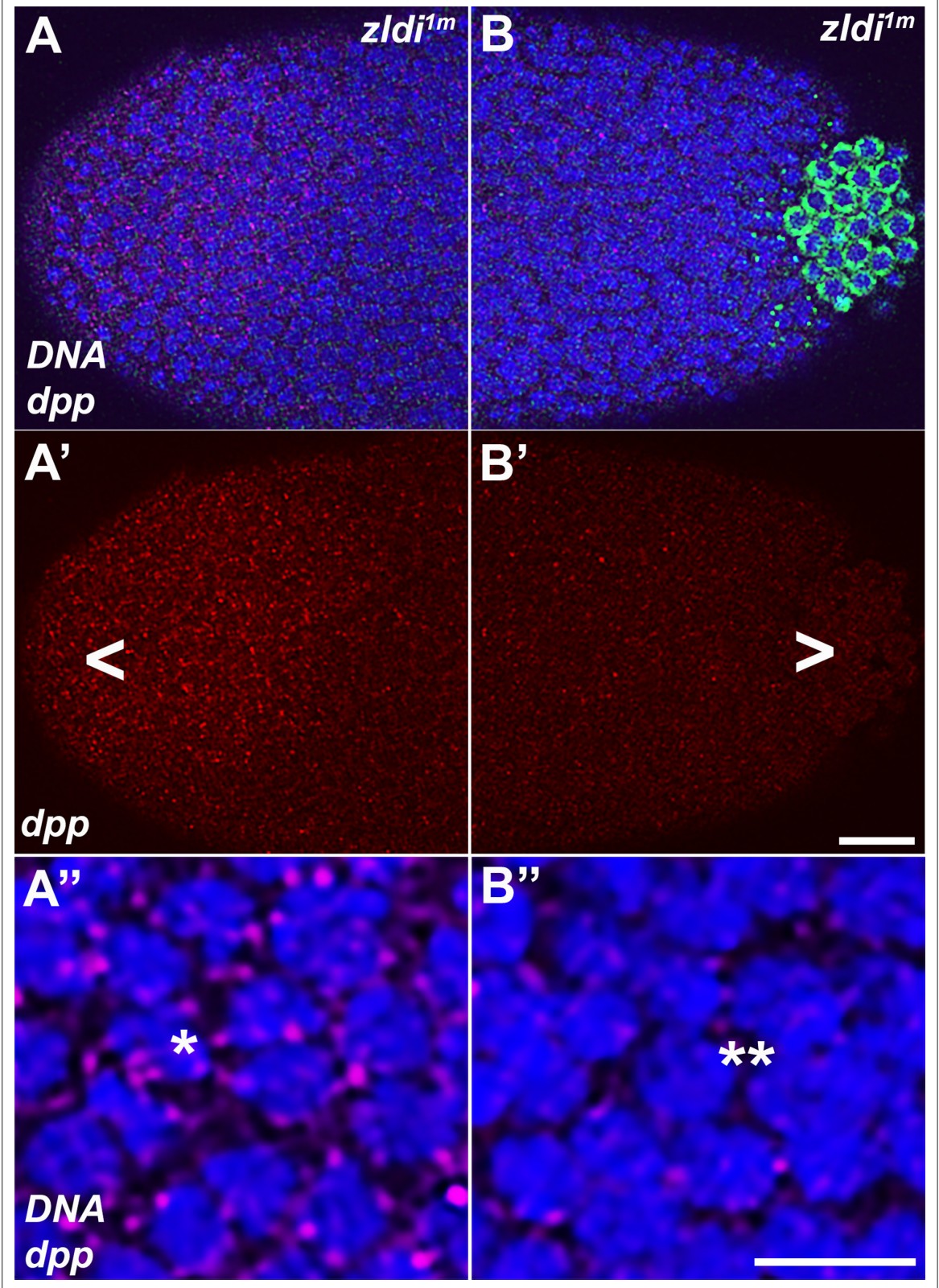

**Figure 14.** Spreading of germ plasm RNAs correlates with enhanced reduction in *decapentaplegic (dpp)* transcripts in the posterior half. Single molecule fluorescence in situ hybridization (smFISH) was performed using probes specific for *dpp* (red, **A–A″** and **B–B″**) *and polar granule component (pgc)* (green, **A and B**) on 0- to 4-hr-old *zldi^1m* embryos to assess the status of somatic transcription at the anterior and posterior poles. While *zldi^1m* embryos express *dpp* (red) at the anterior (**A–A″**), these expression levels are compromised in the posterior half (**B–B″**) as seen by comparison between

*Figure 14 continued on next page*

*Figure 14 continued*

A′ (anterior half, arrowhead) and B′ (posterior half, arrowhead). A″ (anterior) and B″ (posterior) show magnified versions. (Note the difference in signal intensities between nuclei marked with a single asterisk in panel A″ with a double asterisk in B″, respectively.) Scale bar represents 10 µm. Images shown are maximum intensity projections through an embryo at nuclear cycle (NC)13 (late syncytial blastoderm) to capture total germ plasm localization and *dpp* expression throughout relevant focal planes (in Z).

not only because of the loss of *zld*, but also because of the inappropriate localization of germ plasm away from the posterior cortex.

While many of these germline/soma phenotypes manifest themselves most clearly toward the end of the syncytial blastoderm stage or beginning of cellularization (NC13 and NC14, respectively), the likely primary cause—the failure to capture the germline determinants during PGC cellularization—happens earlier during NC9/10 (if not before). Since this is prior to the major wave, it would suggest that ZGA-dependent functions that take place during the minor wave are important for proper PGC cellularization whereby the germline determinants are efficiently sequestered. Most of the phenotypic consequences induced by loss of either Zld or CLAMP proteins can be correlated with dysregulation of centrosome function which is essential for proper PGC determination (*Raff and Glover, 1989*; *Lerit and Gavis, 2011*; *Lerit et al., 2017*; *Colonnetta et al., 2021a*). Previous studies have shown that disruptions in centrosome functions not only cause defects in PGC cellularization, but also lead to the inefficient incorporation of germline determinants into PGCs when they cellularize (*Lerit et al., 2017*; *Colonnetta et al., 2021b*). One plausible idea is that the centrosomal defects in RNAi knockdown embryos are due to the disruption of some critical chromosomal function of the ZGA regulators. In this model, the probable culprit is the striking and pervasive defects in the nuclear division cycles. Nuclear division defects are already evident in *zldi1m* embryos in NC7-9 in the period leading up to the formation of pole buds. Like GAF (*Tsukiyama et al., 1994*; *Wall et al., 1995*), Zld and CLAMP are thought to participate in the remodeling of chromatin, generating regions of DNA that are nucleosome free and accessible for interactions with a variety of DNA binding proteins that have functions not directly related to transcription. These would include components of the origin recognition complex (ORC) and topoisomerase II. The former is responsible for generating the closely spaced replication origins that are essential for completing DNA synthesis in minutes rather hours during the rapid nuclear division cycles. If ORC cannot gain access to the normal complement of replication origins in early embryos, replication will not be completed in time for nuclear division (*Miotto et al., 2016*). The latter, topoisomerase II, functions in decatenating chromosomes after replication is complete (*Nielsen et al., 2020*). Like many other proteins that interact with DNA, topoisomerase II requires open regions of chromatin in order to access the DNA (*Sperling et al., 2011*; *Udvardy and Schedl, 1991*). For this reason, reduction in DNA accessibility in *zld* or *clamp* RNAi knockdown

**Table 3.** Summary of aberrant germline/somatic developmental phenotypes observed in *zld* knockdown embryos.

| Phenotype | Unit classified | Phenotypic frequency in each genotype | | |
|---|---|---|---|---|
| | | *egfpi^m* | *zldi^1m* | *zldi^2m* |
| Germ plasm spread | Embryos | 12/99 (12.1%) | 63/110 (57.3%) | 13/23 (56.5%) |
| Pre-syncytial blastoderm somatic centrosome defects | Nuclei | 3/51 (5.9%) | 13/37 (35.1%) | ND |
| Syncytial blastoderm somatic centrosome defects | Embryos | 2/15 (13.3%) | 15/18 (8.3%) | ND |
| Pole bud/PGC centrosome defects | Nuclei | 4/31 (12.9%) | 16/30 (53.3%) | ND |
| Anterior protrusions | Embryos | 1/12 (8.3%) | 11/18 (61.1%) | ND |
| *slam* expression in PGCs | Embryos | 0/30 (0%) | 2/15 (13.3%) | 1/14 (7.1%) |
| *Sxl-Pe* expression in PGCs | Embryos | 0/30 (0%) | 5/31 (16.1%) | 0/14 (0%) |
| *tll* somatic transcription levels | Embryos | 0/10 (0%) | 8/11 (72.7%) | ND |
| *dpp* somatic transcription levels | Embryos | 0/11 (0%) | 10/13 (76.9%) | ND |

embryos would be expected to impact decatenation. If mitosis proceeds without completing replication and/or decatenation, chromosomes would be expected to fragment, as is observed. This could, in turn, lead to deficiencies in centrosome function, including assembly, duplication, and separation. These centrosomal defects would be expected to interfere with the efficient incorporation of germline determinants into PGCs as they cellularize (*Lerit et al., 2017*; *Lerit and Gavis, 2011*; *Raff and Glover, 1989*). It is also possible that they could trigger the aberrant release and spreading of germ plasm constituents that is observed in *zldi[1m]* knockdown embryos. At this point, it is not clear whether the early mitotic defects observed in embryos compromised for *zelda* activity (*Staudt et al., 2006*, this study) can be tied directly to the failure to establish a sufficient number of nucleosome-free regions or an early deficit in global transcription. Some authors have suggested that transcription in pre-syncytial embryos may be needed to prepare the genome for ZGA (*Ali-Murthy et al., 2013*; *Harrison et al., 2010*), and there could be a similar requirement for the functioning of factors that are important for replication and mitosis.

An alternative (or additional) model is that one or more genes expressed during the minor wave could be important in directing the process of PGC cellularization and normal centrosome function. This is a plausible mechanism as previous studies have shown that PGC formation and proper specification in early embryos depends upon zygotic activity of the BMP pathway (*Colonnetta et al., 2022*). For example, when *dpp* expression is inhibited by zygotic RNAi knockdown, germline determinants like *gcl* and *pgc* mRNAs are not fully captured by the PGCs when they cellularize. Once the PGCs cellularize, a variety of other phenotypes are observed in the *dpp* knockdown. Consistent with this model, *dpp* expression is reduced embryo-wide when *zld* activity is compromised. Moreover, the effects of *zld* knockdown on *dpp* transcription are further exacerbated at the posterior by the spreading of the germ plasm. This in turn would further compromise PGC specification. The BMP pathway is not the only possible transcriptional target. The expression of factors that have a direct role in the assembly or functioning of centrosomes might also be downregulated in *zld* and *clamp* RNAi knockdown embryos.

In sum, together these observations document unprecedented activities of ZGA pioneer factors, Zld and CLAMP. Our data show that the lineage potential of embryonic nuclei is gradually restricted via the combined action of ZGA components as these proteins influence acquisition of either somatic or germline identity during early syncytial blastoderm stages. Thus far, germline identity is thought to be solely determined by Oskar-dependent assembly of germline specific proteins and RNAs. Somatic fate, by contrast, can be essentially viewed as a 'default' state that depends on the absence of germline determinants. Our data argue that ZGA, a process that confers specific fate on individual somatic cells, initially guards their somatic identity. Conversely, by ensuring proper sequestration of the germ plasm, it also protects germline from acquiring a partial somatic fate. Future experiments will focus on how this distinction is set up by the components of ZGA at a molecular level.

# Materials and methods

**Key resources table**

| Reagent type (species) or resource | Designation | Source or reference | Identifiers | Additional information |
|---|---|---|---|---|
| Genetic reagent (*Drosophila melanogaster*) | *Maternal-tubulin-GAL4 (67.15)* | Eric Wieschaus | | |
| Genetic reagent (*Drosophila melanogaster*) | *UAS-egfp RNAi* | Bloomington Drosophila Stock Center | BDSC: 41552; RRID:BDSC_41552 | |
| Genetic reagent (*Drosophila melanogaster*) | *UAS-zld-shRNA (zldi[1])* | Christine Rushlow | | Maintained in the lab of C Rushlow |
| Genetic reagent (*Drosophila melanogaster*) | *UAS-zld RNAi (zldi[2])* | Bloomington Drosophila Stock Center | BDSC: 42016; RRID:BDSC_42016 | |
| Genetic reagent (*Drosophila melanogaster*) | *UAS-clamp RNAi* | Bloomington Drosophila Stock Center | BDSC: 27080; RRID:BDSC_27080 | |
| Antibody | Anti-Vasa (rat polyclonal) | Paul Lasko | RRID:AB_2568498 | Used 1:1000 |

*Continued on next page*

*Continued*

| Reagent type (species) or resource | Designation | Source or reference | Identifiers | Additional information |
|---|---|---|---|---|
| Antibody | Anti-Vasa (mouse monoclonal) | Developmental Studies Hybridoma Bank | DSHB: 46F11; RRID:AB_10571464 | Used 1:10 |
| Antibody | Anti-Cnn (rabbit polyclonal) | Thomas Kaufman | | Used 1:500 |
| Antibody | Anti-Pnut (mouse monoclonal) | Developmental Studies Hybridoma Bank | DSHB: 4C9H4; RRID: AB_528429 | Used 1:10 |
| Sequence-based reagent | *pgc* | *Eagle et al., 2018* | smFISH probe set | Exonic probes |
| Sequence-based reagent | *gcl* | *Eagle et al., 2018* | smFISH probe set | Exonic probes |
| Sequence-based reagent | *osk* | *Little et al., 2015* | smFISH probe set | Exonic probes |
| Sequence-based reagent | *Sxl-Pe* | Thomas Gregor | smFISH probe set | Intronic probes |
| Sequence-based reagent | *slam* | *Colonnetta et al., 2022* | smFISH probe set | Exonic probes |
| Sequence-based reagent | *tll* | *Colonnetta et al., 2021b*; *Colonnetta et al., 2022* | smFISH probe set | Exonic probes |
| Sequence-based reagent | *dpp* | This paper; Biosearch Technologies | smFISH probe set | Exonic probes; sequences available in *Supplementary file 1* |
| Other | Hoescht | Invitrogen | Thermo Fisher Scientific: H3570 | Nuclear dye |

## Fly stocks and genetics

The following *D. melanogaster* stocks were used: *maternal-tubulin-Gal4* line *67.15* (gift from Eric Wieschaus), *zld* shRNA (referred to as *zldi*[1], gift of Christine Rushlow), and *egfp* RNAi (41552), *clamp* RNAi (27080), and *zld* RNAi (referred to as *zldi*[2], 42016) from Bloomington Drosophila Stock Center.

To generate maternal knockdown embryos, we mated *67.15* virgins to males carrying RNAi transgenes, then collected female progeny (all of which carry two copies of *maternal-tubulin Gal4* and one copy of indicated RNAi transgene) that were mated to WT males. The embryos from this cross were then analyzed as 'maternal knockdown' embryos, indicated with [m]. To generate zygotic knockdown embryos, we mated *67.15* virgins to males carrying RNAi transgenes, indicated with [z].

## Immunostaining and single molecule fluorescence in situ hybridization

Embryos were formaldehyde-fixed, and a standard immunohistochemical protocol was used for either DAB or fluorescent staining as described previously (*Deshpande et al., 1999*). Fluorescent immunostaining employed fluorescently labeled (Alexa Fluor) secondary antibodies. The primary antibodies used were mouse anti-Vasa (1:10, DSHB, Iowa City, IA) rat anti-Vasa (1:1000, gift of Paul Lasko), rabbit anti-Centrosomin (1:500, gift from Thomas Kaufmann), and mouse anti-Peanut (1:10, DSHB, Iowa City, IA). Fluorescent immunostaining employed fluorescently labeled (Alexa Fluor) secondary antibodies, used at 1:500 (Thermo Fisher Scientific, Waltham, MA). Embryos were co-labeled with Hoescht (3 µg/ml, Invitrogen, Carlsbad, CA) to visualize nuclei. DAB staining employed secondary anti-peroxidase antibodies (Jackson Immunoresearch Laboratories). Stained embryos were mounted using Aqua Poly/mount (Polysciences, Warrington, PA) on slides and imaged as described below. At least three independent biological replicates were used for each experiment.

Single molecule fluorescence in situ hybridization (smFISH) was performed as described by Little and Gregor using formaldehyde-fixed embryos (*Little et al., 2015*; *Little and Gregor, 2018*). All probe sets were designed using the Stellaris probe designer (20-nucleotide oligonucleotides with 2-nucleotide spacing). *osk*, *pgc*, and *gcl* smFISH probes (coupled to either atto565 or atto647 dye, Sigma, St Louis, MO) were a gift from Liz Gavis (*Little et al., 2015*; *Eagle et al., 2018*), and *Sxl* intronic probes (coupled to either atto565 or atto633 dye, Sigma, St Louis, MO) were a gift from Thomas Gregor (*Colonnetta et al., 2021a*). *tll* probes (coupled to Quasar 570) (*Colonnetta et al., 2021b*), *slam* probes (coupled to Quasar 670) (*Colonnetta et al., 2022*), and *dpp* intronic probes (coupled to

Quasar 670) were produced by Biosearch Technologies (Middlesex, UK). All samples were mounted using Aqua Poly/mount (Polysciences, Warrington, PA) on slides and imaged as described below. At least three independent biological replicates were used for each experiment.

## Microscopy and image processing

NIKON-Microphot-SA microscope was used to image and analyze DAB-stained embryos. Confocal imaging for all fluorescently labeled samples was performed on a Nikon A1 inverted laser-scanning confocal microscope. Unless noted in figure legend, all images were of single sections, not maximum intensity projections from Z stacks. To assess the spreading of the RNAs or protein in different mutant backgrounds compared to the control, we generated plot profiles using ImageJ. The posterior-most 75 µm of each embryo was plotted for comparison, and embryos from a single biological replicate are plotted in figures given that variation between fluorescence between replicates obscured the germ plasm distribution trends if embryos from all replicates were plotted together. Images were assembled using Fiji (ImageJ, NIH) and Adobe Photoshop software to crop regions of interest, adjust brightness and contrast, and separate or merge channels.

## Classifying embryonic germline/somatic distinction phenotypes

Typically, we analyzed embryonic phenotypes by classifying each image blindly, unaware of which genotype was represented in the image. We also determined 'PGC' nuclei versus 'somatic' nuclei based on location—PGC or pole bud nuclei are located at the posterior cap of an embryo while somatic nuclei include the remainder of the developing embryonic syncytium.

PGCs in control or WT embryos display uniformly high levels of Vasa while experimental embryos (*zldi1m*, for instance) have variable levels of Vasa among PGCs, which also show decreased PGC numbers and aberrant spacing/localization. Therefore, we classified 'low' versus 'high' levels of Vasa in PGCs for each blindly scored embryo. Likewise, we scored embryos for 'spread' versus 'no spread' of germ plasm RNAs, based on the localization of germ plasm RNAs at or away from the posterior terminus of the embryo. We then compiled all data points for embryos within each genotype for statistical analysis (further detailed below).

## Statistical analysis

Using NC13/14 embryos, PGCs of each genotype were counted from the first Vasa-positive cell to the last through an entire z-volume captured at 2 µm intervals. These PGCs counts were analyzed using a Student's t-test. These PGCs were also classified as having high or low levels of Vasa, and pairwise comparisons of these populations for each genotype were performed using Fisher's exact test. Likewise, proportions of embryos displaying aberrant germ plasm localization (degree of spreading) were compared to control embryos using Fisher's exact test. To analyze the effect of NC progression on the germ plasm spreading phenotype, we used ordinal logistic regression, regressing stage (classified as pre-blastoderm, early syncytial blastoderm, late syncytial blastoderm, or cellular blastoderm), and genotype on degree of germ plasm spread. For smFISH experiments, total number of embryos expressing *slam* or *Sxl-Pe* in PGCs were counted, and Fisher's exact test was used to test significance in the compared proportions of embryos positive for transcription in PGCs.

Data were plotted and statistical analyses were performed using Microsoft Excel and R Project software.

## Acknowledgements

Authors gratefully acknowledge Eric Wieschaus and Trudi Schupbach for many discussions, useful suggestions, and reagents over the course of this work. Mike Levine, Liz Gavis, Stas Shvartsman, and Shelby Blythe are thanked for continued support. Chris Ng and Gordon Grey provided technical assistance and fly food, respectively. We thank Dr Gary Laevsky and the Confocal Imaging Facility, a Nikon Center of Excellence, in the Department of Molecular Biology at Princeton University for instrument use. This work was supported by grants from National Institute of Health (NICHD:093913) to PS and GD, and (NIGMS: 126975) to PS. MC was supported by NSF Graduate Research Fellowship (DGE-1656466).

## Additional information

### Funding

| Funder | Grant reference number | Author |
|---|---|---|
| National Institutes of Health | GM126975 | Girish Deshpande |
| National Institutes of Health | HD093913 | Girish Deshpande |

The funders had no role in study design, data collection and interpretation, or the decision to submit the work for publication.

### Author contributions

Megan M Colonnetta, Conceptualization, Data curation, Formal analysis, Funding acquisition, Validation, Investigation, Writing - original draft, Writing – review and editing; Paul Schedl, Conceptualization, Funding acquisition, Project administration, Writing – review and editing; Girish Deshpande, Conceptualization, Formal analysis, Supervision, Funding acquisition, Validation, Investigation, Writing - original draft, Project administration, Writing – review and editing

### Author ORCIDs

Megan M Colonnetta (ID) http://orcid.org/0000-0001-5685-1670
Girish Deshpande (ID) http://orcid.org/0000-0002-5200-7090

### Decision letter and Author response

Decision letter https://doi.org/10.7554/eLife.78188.sa1
Author response https://doi.org/10.7554/eLife.78188.sa2

## Additional files

### Supplementary files

• Supplementary file 1. *Decapentaplegic (dpp)* exonic single molecule fluorescence in situ hybridization (smFISH) probe sequences.

• Transparent reporting form

### Data availability

All data generated or analyzed in this study are included in the manuscript and supporting file; Source data files have been provided for for relevant figures.

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
