## [Editor Report]

The early differentiation of germ cells, those that will form egg and sperm, is a critical and nearly universal step in animal development. This paper reveals new layers of molecular and cellular regulation that control this process in the fly, and as such be of broad interest to cell and developmental biologists, especially those interested in critical cell fate decisions. The paper contains a wealth of experimental data demonstrating that processes generally thought to be restricted to somatic cells alter the differentiation of germ cells, but provides only limited functional interpretation of the observed phenotypes.

---

## [Decision Letter]

**Decision letter after peer review:**

Thank you for submitting your article "Germline/soma distinction in *Drosophila* embryos requires regulators of zygotic genome activation" for consideration by *eLife*. Your article has been reviewed by 2 peer reviewers, and the evaluation has been overseen by a Reviewing Editor and Michael Eisen as the Senior Editor. The reviewers have opted to remain anonymous.

Essential revisions:

Please see detailed reviews for specifics.

1) The paper is rich in description but limited in its functional interpretation. Discussion of the phenotypes (e.g. centrosome defects, germplasm locations defects, abnormal gene expression) and how they related to each other and more broadly to PGC differentiation is essential.

2) As the paper relies on the quantification of microscopy data, it is essential that the revised version provide a clearer explanation of and justification for how this quantification was done.

3) The reviewers raised concerns about whether the maternal and zygotic RNAi knockdowns were specific to these RNA pools (especially whether the maternal knockdown also affected zygotic transcripts). The specificity either needs to be addressed with experimental data, or clarified in some other manner.

4) The reviewers have some specific issues/suggestions for figures that should be addressed.

*Reviewer #1 (Recommendations for the authors):*

1) The authors show multiple defects in Zelda/CLAMP mutants, such as 1. centrosome defect in soma, 2. centrosome defect in PGC, 3. Germplasm localization defect, 4. Abnormal gene expression in soma and PGC, 5. dpp expression. The authors need to state their conclusion or discussion how these phenotypes related each other.

2) The experiment of comparison of maternal knock and zygotic knock down has to be designed better way as maternal Gal4 will be deposited to embryo to knock down zygotic gene. Does zygotic Zelda expression in the background of zelda mutant may be useful to strengthen this point.

3) Scoring criteria (detailed methods and threshold if they set any) throughout most figures has to be stated thoroughly. For example, how did they use any threshold to judge "reduced level of Vasa" vs "normal level of Vasa" in Figure1? Similarly, how did they determine "spread" vs "not spread" of pgc mRNA in Figure2?

*Reviewer #2 (Recommendations for the authors):*

1) The reported findings are based entirely on imaging of proteins by immunostaining (Vasa, Cnn, Pnut) and RNAs by smFISH (pgc, gcl, oskar, slam, Sxl-Pe, tll, dpp). The images for the most part present clear evidence of the defects claimed by the authors. And the distribution diagrams in Figure 2 and 3 display the range of defects in embryos. A significant shortcoming of some of the analyses is that, although the authors reported on the percentages of embryos that displayed particular defects, it is not clearly explained how the images were quantified to arrive at those percentages. , such as quantifying levels of Vasa staining, counting smFISH foci in regions of interest, and assessing whether few or many of the relevant cells (e.g. pole cells) in embryos displayed defects. If such quantification was part of the analyses, it should be explained better. If not, either it should be done or else the authors should explain their scoring rubric. For example, would 1 smFISH dot of slam or Sxl-Pe in a pole cell lead to classification of that embryo as defective? Some of the images would also benefit from circumscribing the PGCs to make clear which are PGCs and which are nearby somatic nuclei/cells.

2) In several places, the authors compared the defects caused by loss of maternal versus zygotic Zelda or CLAMP. Loss of maternal (m-) used mothers that expressed an RNAi or shRNA transgene during oogenesis. Loss of zygotic (z-) used fathers that delivered an RNAi transgene to embryos. I think that in general m- and z- caused similar defects, which led me to wonder if knock-down of Zelda or CLAMP in oocytes in fact knocks down both maternal and zygotic. In other words, would RNAi or shRNA driven in the maternal germline be loaded into embryos to knock down zygotic expression as well? If maternal knock-down (m-) is actually maternal plus zygotic knock-down (m-z-), then comparing the effects of m- and z- is actually comparing m-z- and z- and is not valid.

3) What stages do 0-4 hr old embryos correspond to? The figure legends should include the stages of the embryos shown (e.g. early syncytial blastoderm, late syncytial blastoderm, or cellular blastoderm) based on nuclear density as described on p. 13 and shown in Table 2?

4) Figure 9 and 10: The anterior pole-cell-like budding phenomenon is really interesting. It sounds as if that occurs without any of the critical pole plasm components at the anterior pole. If that is right, then the implication is that the anterior end has the potential for pole-cell-like budding and that ZGA normally turns on a gene that blocks that. Were live embryos examined for anterior budding, as I imagine that is transient and might not be efficiently captured by fixing and staining, leading to an underestimation of the % of embryos that show anterior budding?

5) Figure 12: This figure should include a control embryo showing elevated levels of tll transcripts uniformly in the anterior and posterior, as mentioned in the text.

---

## [Author Response]

Essential revisions:Please see detailed reviews for specifics.1) The paper is rich in description but limited in its functional interpretation. Discussion of the phenotypes (e.g. centrosome defects, germplasm locations defects, abnormal gene expression) and how they related to each other and more broadly to PGC differentiation is essential.

This is an important point, and we recognize that the earlier version of Discussion only hinted at possibilities but lacked a description of a mechanistic scenario connecting the dots. As recommended, we have now reworked the discussion to emphasize that the likely cause of the problems in PGC vs soma specification are defects in centrosomal functions. We also discuss mechanisms that might give rise to the centrosmal defects in the RNAi knockdown embryos.

2) As the paper relies on the quantification of microscopy data, it is essential that the revised version provide a clearer explanation of and justification for how this quantification was done.

We have tried to improve this description in several places. We have included a brief description of how embryos were classified in the Methods section of the revised manuscript (lines 510-522). As recommended, we have also included a table showing how many embryos/nuclei were analyzed and what proportion of embryos/nuclei displayed the phenotypes (Table 3 in Discussion). These phenotypes were used as a basis for most of the quantifications and calculation of significance values.

We have also provided some additional examples to illustrate our point in response to point number 3 of reviewer (1). (We have not included these in the manuscript, however).

3) The reviewers raised concerns about whether the maternal and zygotic RNAi knockdowns were specific to these RNA pools (especially whether the maternal knockdown also affected zygotic transcripts). The specificity either needs to be addressed with experimental data, or clarified in some other manner.

We agree with the reviewers, and therefore we have modified the text acknowledging the likelihood that maternal RNAi based knockdown (KD) also impacts zygotic gene expression. In labeling the knockdowns as “maternal” and “zygotic” we were indicating at which stage the RNAi is first expressed. For zygotic KD, this is straightforward: The mother provides the Gal4 protein while the father provides the UAS RNAi transgene. The maternal KD is obviously more complicated. For one, the processed dsRNAs are likely deposited in the egg and thus could persist and remain active in the embryos. Likewise, in embryos inheriting the UAS RNAi transgene, there could be zygotic expression.

The maternal RNAi knockdown was used as we were unable to obtain a sufficient number of embryos from *zelda* germline clones. However, Christine Rushlow’s lab has reported that Zelda RNAi embryos resemble the Zelda m^-^ embryos derived from females carrying germline clones as RNAseq analysis of germline clone vs. maternal KD embryos yielded similar targets. In the case of CLAMP, we found that RNAi knockdown of CLAMP in the mother induced oogenesis defects including the mislocalization of *orb* mRNA. As we wanted to examine the effects of CLAMP knockdown during embryogenesis (without complications arising from oogenesis defects) we used only zygotic knockdowns for CLAMP.

Importantly, for the Zelda knockdowns, we observed a similar set of phenotypic consequences for both the “maternal” and zygotic knockdowns (though the defects were less severe in the zygotic knockdowns as would be expected).

4) The reviewers have some specific issues/suggestions for figures that should be addressed.

We have tried to modify the text and figures as suggested to address most of these points. We have also included a detailed response to the specific points raised by the reviewers. In our assessment, addressing these issues has substantially improved the quality of the manuscript and hope that the revised version will be suitable for publication in *eLife*.

Reviewer #1 (Recommendations for the authors):1) The authors show multiple defects in Zelda/CLAMP mutants, such as 1. centrosome defect in soma, 2. centrosome defect in PGC, 3. Germplasm localization defect, 4. Abnormal gene expression in soma and PGC, 5. dpp expression. The authors need to state their conclusion or discussion how these phenotypes related each other.

This is a useful suggestion and we have reworked the discussion focusing on possible mechanistic connections between the different phenotypes.

2) The experiment of comparison of maternal knock and zygotic knock down has to be designed better way as maternal Gal4 will be deposited to embryo to knock down zygotic gene. Does zygotic Zelda expression in the background of zelda mutant may be useful to strengthen this point.

Please see our response to major point 3 in our letter to the editor (above).

3) Scoring criteria (detailed methods and threshold if they set any) throughout most figures has to be stated thoroughly. For example, how did they use any threshold to judge "reduced level of Vasa" vs "normal level of Vasa" in Figure1? Similarly, how did they determine "spread" vs "not spread" of pgc mRNA in Figure2?

We analyzed these phenotypes by classifying each image blindly, unaware of which genotype was represented in the image. As shown in Author response image 1, PGCs in control or WT embryos display uniformly high levels of Vasa while experimental embryos (*zldi^1m^*, for instance) have variable levels of Vasa among PGCs, which also show decreased PGC numbers and aberrant spacing/localization. Therefore, we classified “low” vs. “high” levels of Vasa in PGCs for each blindly scored embryo. Likewise, we scored embryos for “spread” vs. “no spread” of germ plasm RNAs, as exhibited in the examples in Author response image 2 (control/no spread, intermediate vs. strong spreading/mislocalization of *pgc*, a germ plasm RNAs). We then compiled all data points for embryos within each genotype for statistical analysis. Also, in addition to overt reduction in the total number of pole cells and mislocalization of the pole plasm components, we observed other interesting phenotypes. For instance, we noticed gaps between newly formed pole cells whereas PGCs are usually present in a continuous monolayer. Furthermore, at times PGC-like, i.e. Vasa- positive cells were formed away from the extreme posterior. These phenotypes are illustrated in Author response image 2 in *zldi^1m^* embryos. While these phenotypes are not highly penetrant, they are *never* observed in control samples and thus worth mentioning. Together these data strongly argue in favor of an important function for the ZGA components during early embryonic germline/soma distinction. We hope to uncover the mechanistic underpinnings of this novel activity.

We have added a short paragraph explaining these details in the Methods section in the revised version (lines 510-522).

**Author response image 1. sa2fig1:** 

Reviewer #2 (Recommendations for the authors):1) The reported findings are based entirely on imaging of proteins by immunostaining (Vasa, Cnn, Pnut) and RNAs by smFISH (pgc, gcl, oskar, slam, Sxl-Pe, tll, dpp). The images for the most part present clear evidence of the defects claimed by the authors. And the distribution diagrams in Figure 2 and 3 display the range of defects in embryos. A significant shortcoming of some of the analyses is that, although the authors reported on the percentages of embryos that displayed particular defects, it is not clearly explained how the images were quantified to arrive at those percentages. , such as quantifying levels of Vasa staining, counting smFISH foci in regions of interest, and assessing whether few or many of the relevant cells (e.g. pole cells) in embryos displayed defects. If such quantification was part of the analyses, it should be explained better. If not, either it should be done or else the authors should explain their scoring rubric. For example, would 1 smFISH dot of slam or Sxl-Pe in a pole cell lead to classification of that embryo as defective? Some of the images would also benefit from circumscribing the PGCs to make clear which are PGCs and which are nearby somatic nuclei/cells.

We thank the reviewer for pointing this out. It should be noted that many of the reported observations initially took us by surprise. In fact, earlier results from the Rushlow lab explicitly stated this in their early report. As ZGA components were not supposed to influence germ cell fate, we analyzed the data by classifying the embryos in a qualitative manner in our initial experiments. Nevertheless, the differences were clearcut. Subsequently we used >10 embryos of respective genotypes to ascribe each embryo to one of three categories: Near wild type, intermediate, or severe. To avoid confusion, we did not consider mild defects. Subsequently, we employed line traces to assess if distribution of the pole plasm RNAs appears different compared to the control population. These data, in our opinion, revealed clear differences supporting our model.

The PGCs are transcriptionally quiescent and somatic genes such as *ftz*/*eve*, *Sxl,* and *slam* are never transcribed in wild type (*w^1^* or *Ore-R*) or control (*egfp-RNAi*) PGCs. We have previously shown that in embryos maternally compromised for *polar granule component* (*pgc*), *nanos* (*nos*) or *germ cell less* (*gcl*), some but not all of these genes are ectopically transcribed in the PGCs (Deshpande *et al.* 1999, Deshpande *et al.* 2004, Martinho *et al.* 2004, Colonnetta *et al.* 2021). As we observed that these germ plasm RNAs are not anchored properly to the posterior cortex of embryos maternally compromised for ZGA components, we also sought to test if a subset of the PGCs ectopically activated transcription of some of the somatic genes. As the localization defects in any given embryo are only partially penetrant, we did not expect the phenotypic consequences to be severe. As anticipated, we observed ectopic transcription in a small number of ‘mutant’ PGCs which was statistically significant as compared to the control PGCs. Also, as noted previously, most of these genes are targets of ZGA regulators and are downregulated in the soma upon loss of Zelda or CLAMP activity. Curiously, however, a subset of pole cells display some of these transcripts. So, while scoring, we were able to score not only the ectopic presence of transcripts in the PGCs but also a reduction in the soma of the same transcript compared to the control embryos. This type of reciprocal behavior i.e. reduction in the somatic nuclei while sporadic increase in the PGCs, in fact, served as a good internal control.

2) In several places, the authors compared the defects caused by loss of maternal versus zygotic Zelda or CLAMP. Loss of maternal (m-) used mothers that expressed an RNAi or shRNA transgene during oogenesis. Loss of zygotic (z-) used fathers that delivered an RNAi transgene to embryos. I think that in general m- and z- caused similar defects, which led me to wonder if knock-down of Zelda or CLAMP in oocytes in fact knocks down both maternal and zygotic. In other words, would RNAi or shRNA driven in the maternal germline be loaded into embryos to knock down zygotic expression as well? If maternal knock-down (m-) is actually maternal plus zygotic knock-down (m-z-), then comparing the effects of m- and z- is actually comparing m-z- and z- and is not valid.

We completely agree and have changed the description to acknowledge this possibility. As stated earlier we were primarily interested in evaluating whether just zygotic KD can recapitulate the important aspects of the phenotypic consequences induced by the maternal KD. This is indeed the case.

3) What stages do 0-4 hr old embryos correspond to? The figure legends should include the stages of the embryos shown (e.g. early syncytial blastoderm, late syncytial blastoderm, or cellular blastoderm) based on nuclear density as described on p. 13 and shown in Table 2?

Thank you for this suggestion. This information has been included in each figure legend.

4) Figure 9 and 10: The anterior pole-cell-like budding phenomenon is really interesting. It sounds as if that occurs without any of the critical pole plasm components at the anterior pole. If that is right, then the implication is that the anterior end has the potential for pole-cell-like budding and that ZGA normally turns on a gene that blocks that. Were live embryos examined for anterior budding, as I imagine that is transient and might not be efficiently captured by fixing and staining, leading to an underestimation of the % of embryos that show anterior budding?

We appreciate the feedback. We are also curious about these protrusions observed in the anterior of the embryos. We agree that the presence of these transient bud-like protrusions at the anterior end can't be readily attributed to the pole plasm spreading. However minor traces of pole plasm at the wrong end could contribute to such cell biological aberrations and as suggested live imaging may help shed some light on the dynamic cytoskeletal changes that lead to these protrusions. We would like to point out that such anterior budding was previously observed in *eve^-^* mutant embryos by Tom Kornberg and colleagues (Ali-Murthy *et al.* 2013). Therefore, it seems possible that aberration in zygotic transcription during the minor phase of ZGA induced by loss of Zelda/CLAMP could result in ectopic protrusions. Alternatively changes in centrosome behavior can induce such ectopic budding. (Moreover, these two possibilities are not mutually exclusive). Our data suggests that the somatic centrosomes are indeed abnormal in the embryos compromised for Zelda or CLAMP. Future analysis will thus help reveal the mechanistic underpinnings of this interesting observation.

5) Figure 12: This figure should include a control embryo showing elevated levels of tll transcripts uniformly in the anterior and posterior, as mentioned in the text.

We have now included an appropriate control embryo in a supplementary figure (Figure 12 —figure supplement 1).